# Effective radiative forcing and adjustments in CMIP6 models

Christopher J. Smith[1,2], Ryan J. Kramer[3,4], Gunnar Myhre[5], Kari Alterskjær[5], William Collins[6], Adriana Sima[7], Olivier Boucher[8], Jean-Louis Dufresne[7], Pierre Nabat[9], Martine Michou[9], Seiji Yukimoto[10], Jason Cole[11], David Paynter[12], Hideo Shiogama[13,14], Fiona M. O'Connor[15], Eddy Robertson[15], Andy Wiltshire[15], Timothy Andrews[15], Cécile Hannay[16], Ron Miller[17], Larissa Nazarenko[17], Alf Kirkevåg[18], Dirk Olivié[18], Stephanie Fiedler[19], Anna Lewinschal[20], Chloe Mackallah[21], Martin Dix[21], Robert Pincus[22,23], and Piers M. Forster[1]

[1]School of Earth & Environment, University of Leeds, LS2 9JT, UK
[2]International Institute for Applied Systems Analysis (IIASA), Laxenburg A-2361, Austria
[3]Climate and Radiation Laboratory, NASA Goddard Space Flight Center, Greenbelt, MD 20771, USA
[4]Universities Space Research Association, 7178 Columbia Gateway Drive, Columbia, MD 21046, USA
[5]CICERO, Oslo, Norway
[6]Department of Meteorology, University of Reading, UK
[7]LMD/IPSL, Sorbonne Université, ENS, PSL Université, École polytechnique, Institut Polytechnique de Paris, CNRS, Paris France
[8]Institut Pierre-Simon Laplace, Sorbonne Université / CNRS, Paris, France
[9]CNRM, Université de Toulouse, Météo-France, CNRS, Toulouse, France
[10]Meteorological Research Institute, Tsukuba, Japan
[11]Canadian Centre for Climate Modelling and Analysis, Environment Canada, Victoria, British Columbia, Canada
[12]Geophysical Fluid Dynamics Laboratory, Princeton University Forrestal Campus, 201 Forrestal Road, Princeton, NJ 08540-6649
[13]Center for Global Environmental Research, National Institute for Environmental Studies, 16-2 Onogawa, Tsukuba, Ibaraki 305-8506, Japan
[14]Atmosphere and Ocean Research Institute, University of Tokyo, 5-1-5 Kashiwanoha, Kashiwa, Chiba 277-8564, Japan
[15]Met Office Hadley Centre, FitzRoy Road, Exeter, EX1 3PB, UK
[16]NCAR/UCAR, Boulder, Colorado, USA
[17]NASA Goddard Institute for Space Studies, New York, NY 10025 USA
[18]Norwegian Meteorological Institute, Oslo, Norway
[19]University of Cologne, Institute of Geophysics and Meteorology, Cologne, Germany
[20]Department of Meteorology, Stockholm University, Sweden
[21]CSIRO Oceans and Atmosphere, Aspendale, Australia
[22]Cooperative Institute for Research in Environmental Sciences, University of Colorado Boulder, CO, USA
[23]NOAA/ESRL Physical Sciences Division, Boulder, CO, USA

**Correspondence:** C.J. Smith (c.j.smith1@leeds.ac.uk)

**Abstract.** The effective radiative forcing, which includes the instantaneous forcing plus adjustments from the atmosphere and surface, has emerged as the key metric of evaluating human and natural influence on the climate. We evaluate effective radiative forcing and adjustments in 17 contemporary climate models that are participating in CMIP6 and have contributed to the Radiative Forcing Model Intercomparison Project (RFMIP). Present-day (2014) global mean anthropogenic forcing relative to pre-industrial (1850) from climate models stands at 2.00 ($\pm$ 0.23) W m$^{-2}$, comprised of 1.81 ($\pm$ 0.09) W m$^{-2}$ from $CO_2$, 1.08 ($\pm$ 0.21) W m$^{-2}$ from other well-mixed greenhouse gases, $-1.01$ ($\pm$ 0.23) W m$^{-2}$ from aerosols and $-0.09$ ($\pm$ 0.13) W

m$^{-2}$ from land use change. Quoted uncertainties are one standard deviation across model best estimates, and 90% confidence in the reported forcings, due to internal variability, is typically within 0.1 W m$^{-2}$. The majority of the remaining 0.21 W m$^{-2}$ is likely to be from ozone. In most cases, the largest contributors to the spread in ERF is from the instantaneous radiative forcing (IRF) and from cloud responses, particularly aerosol-cloud interactions to aerosol forcing. As determined in previous studies, cancellation of tropospheric and surface adjustments means that the stratospherically adjusted radiative forcing is approximately equal to ERF for greenhouse gas forcing, but not for aerosols, and consequentially, not for the anthropogenic total. The spread of aerosol forcing ranges from $-0.63$ to $-1.37$ W m$^{-2}$, exhibiting a less negative mean and narrower range compared to 10 CMIP5 models. The spread in $4{\times}CO_2$ forcing has also narrowed in CMIP6 compared to 13 CMIP5 models. Aerosol forcing is uncorrelated with climate sensitivity. Therefore, there is no evidence to suggest that the increasing spread in climate sensitivity in CMIP6 models, particularly related to high-sensitivity models, is a consequence of a stronger negative present-day aerosol forcing, and little evidence that modelling groups are systematically tuning climate sensitivity or aerosol forcing to recreate observed historical warming.

## 1 Introduction

The effective radiative forcing (ERF) has gained acceptance as the most useful measure of defining the impact on the Earth's energy imbalance to a radiative perturbation (Myhre et al., 2013; Boucher et al., 2013; Forster et al., 2016). These perturbations can be anthropogenic or natural in origin, and include changes in greenhouse gas concentrations, aerosol burdens, land use characteristics, solar activity, and volcanic eruptions. Since the start of the Industrial Era until the present-day, anthropogenic forcing has typically been increasing, and has been the dominant component of the total forcing on the Earth system except for brief periods following large volcanic eruptions (Myhre et al., 2013). The main constituents of anthropogenic ERF are a positive forcing from greenhouse gases and a partially offsetting negative forcing from aerosols. While greenhouse gas forcing is reasonably well-known, aerosol forcing is more uncertain due to the spatial variation of aerosols, their short atmospheric lifetime, and their complex interactions with clouds (Boucher et al., 2013; Bellouin et al., 2020b).

ERF is useful because equilibrium temperatures are more closely related to surface warming in the forcing-feedback relationship of the Earth's atmosphere:

$$\Delta N = F - \lambda \Delta T \tag{1}$$

where $\Delta N$, $F$, $\lambda$ and $\Delta T$ are the top-of-atmosphere (TOA) energy imbalance, (effective) radiative forcing, climate feedback parameter, and change in global-mean surface air temperature respectively. Richardson et al. (2019) showed that using ERF rather than RF reduces the need for forcing-specific efficacy values (the temperature response per unit forcing), first introduced by Hansen et al. (2005) as an observation that different values of $\lambda$ better predicted $\Delta T$ for different forcing agents under

RF. Conversely, evaluating ERF is less straightforward than RF, requiring climate model integrations, and numerous different methods of calculating ERF exist with their own benefits and drawbacks (Shine et al., 2003; Gregory et al., 2004; Hansen et al., 2005; Forster et al., 2016; Tang et al., 2019; Richardson et al., 2019).

The difference between ERF and RF is that ERF includes all tropospheric and land-surface adjustments whereas RF only includes the adjustment due to stratospheric temperature change (Sherwood et al., 2015; Myhre et al., 2013). Adjustments are often termed "rapid" (Myhre et al., 2013; Smith et al., 2018b), however, there is no formal separation of adjustments and climate feedbacks based on timescale alone (Sherwood et al., 2015). It is conceptually more appropriate to divide adjustments as those changes in state that occur purely as a result of the action of a forcing agent from slow feedbacks that occur as a

result of a change in global mean surface temperature. The instantaneous radiative forcing (IRF) is the initial perturbation to the Earth's radiation budget and unlike the RF and ERF does not include adjustments. By analysing atmosphere-only climate simulations using fixed climatological sea-surface temperatures (SSTs) and sea ice distributions, surface temperature driven feedbacks are largely suppressed except for a small contribution from land surface warming or cooling (Vial et al., 2013; Tang et al., 2019), allowing for adjustments to be diagnosed from atmospheric state changes (Forster et al., 2016; Smith et al., 2018b).

This provides insight into the mechanisms contributing to the effective radiative forcing. For example, the ERF of black carbon is half of the impact estimated from its IRF as a consequence of its strong atmospheric absorption and adjustments arising from how it perturbs tropospheric heating rates, affecting the distribution of tropospheric temperatures, water vapour and clouds (Stjern et al., 2017; Smith et al., 2018b; Johnson et al., 2019; Allen et al., 2019).

The experimental protocol for determining (effective) radiative forcing in models has been extended since Phase 5 of the

Coupled Model Intercomparison Project (CMIP5). CMIP5 included experiments for present-day (year 2000) all-aerosol and sulfate-only forcing (Zelinka et al., 2014, CMIP5 experiment labels sstClimAerosol and sstClimSulfate), and $4 \times CO_2$ forcing (sstClim4xCO2; Andrews et al., 2012; Kamae and Watanabe, 2012) with respect to a pre-industrial baseline with climatological SSTs and sea ice distributions (sstClim). A handful of IRF outputs from quadrupled $CO_2$ experiments (Chung and Soden, 2015) were also obtained. For CMIP6, the Radiative Forcing Model Intercomparison Project (RFMIP; Pincus et al., 2016) provides

a number of present-day time-slice and historical-to-future transient experiments designed to evaluate the ERF in climate models for different forcing agents, providing insight into why climate models respond the way they do to particular forcings. This is important when diagnosing climate feedbacks (Forster et al., 2013), given the role of forcing in the Earth's energy budget (eq. (1)), and knowledge of forcing is required for attribution of historical temperature change (Haustein et al., 2017), evaluating non-$CO_2$ contributions to remaining carbon budgets (Tokarska et al., 2018), and in future scenario projections

(Gidden et al., 2019). Effective radiative forcings derived from models can be used to validate assumptions derived from other lines of evidence, particularly for aerosol forcing, as is done by the Intergovernmental Panel on Climate Change (IPCC) in their periodic Assessment Reports.

## 2 Models and experimental protocol

We use results from 17 state-of-the-art atmospheric general circulation models (GCMs) and Earth system models (ESMs) contributing to Tier 1 of RFMIP (table 1) as part of CMIP6 (Eyring et al., 2016). In addition, GISS-E2-1-G provided two physics variants, r1i1p1f1 and r1i1p3f1, with aerosol treatments that are different enough to justify treating the variants as separate models, bringing the total to 18. Models with diagnostics available on the Earth System Grid Foundation (ESGF) up until 13 May 2020 have been analysed. Each model is run in atmosphere-only mode using pre-industrial climatologies of sea-surface temperatures (SSTs) and sea-ice distributions from at least 30 years of the same model's corresponding coupled pre-industrial control run (piControl, Eyring et al. (2016)). RFMIP's Tier 1 calls for 30-year timeslice experiments forced with $4\times$ pre-industrial $CO_2$ concentrations (RFMIP name piClim-4xCO2), all present-day anthropogenic forcers (piClim-anthro), present-day well-mixed greenhouse gases (piClim-ghg), present-day aerosols (piClim-aer) and present-day land use (piClim-lu) in this fixed-SST configuration. All forcing components that are not perturbed in a particular experiment remain at pre-industrial (year 1850) values, and "present-day" is defined as year 2014 conditions. A 30-year experiment with pre-industrial conditions, piClim-control, is also performed as a reference case, and all results presented in this paper are with reference to piClim-control, accounting for the possibility that models may have a non-zero pre-industrial TOA flux imbalance. Results from the $4\times CO_2$ experiment are also rescaled to the ratio of 2014 to 1850 $CO_2$ concentrations of approximately $1.4\times$ pre-industrial by a factor of 0.2266, being the ratio of RF from $1.4\times CO_2$ to $4\times CO_2$ from the Etminan et al. (2016) formula. This is performed to isolate an estimate of the $CO_2$-only contribution to the present-day forcing, and is based on year-1850 and year-2014 $CO_2$ concentrations of 284.32 and 397.55 ppm respectively (Meinshausen et al., 2017) along with the 1850 concentrations of 808.25 ppb for $CH_4$ and 273.02 ppb for $N_2O$. Except where explicitly stated, we present results from this experiment as $1.4\times CO_2$.

The experiments and results presented in this study follow on from the assessment of ERF and adjustments in 11 models contributing to the Precipitation Driver and Response Model Intercomparison Project (PDRMIP, see Myhre et al., 2017) in Smith et al. (2018b). In Smith et al. (2018b) idealised experiments of $2\times CO_2$ concentrations, $3\times CH_4$ concentrations, $10\times$ black carbon (BC) emissions or burdens, $5\times SO_4$ emissions or burdens and a 2% solar constant increase were analysed from CMIP5-era and interim models. Only the $4\times CO_2$ experiment has a similar experiment for comparison in Smith et al. (2018b), whereas the RFMIP protocol focuses more on combinations of anthropogenic forcers. In addition, extended model diagnostics allow us to determine cloud responses and aerosol forcing in more detail in this study.

## 3 Effective radiative forcing

Using climatological SSTs allows for ERF to be diagnosed as the difference of top-of-atmosphere net radiative flux between a given forcing experiment and a pre-industrial control simulation (Hansen et al., 2005). Using 30 year timeslices generally results in standard absolute errors of less than 0.1 W m$^{-2}$ (Forster et al., 2016). Although inter-annual variability affects the diagnosed ERF using this climatological SST method, the standard error in the estimates obtained is much smaller than using a fully-coupled ocean-atmosphere model with a Gregory regression (Gregory et al., 2004), and as such fewer model years are

**Table 1.** Contributing climate models to RFMIP-ERF Tier 1. The adjustment time is based on approximately how long stratospheric temperatures take to equilibrate in the $4\times CO_2$ experiment (fig. 2). ISCCP simulator diagnostics are indicated where existent.

| Model | Atmospheric resolution (lon × lat) | Adjustment timescale (yr) | Model years | ISCCP simulator | Reference |
|---|---|---|---|---|---|
| ACCESS-CM2 | $1.875° \times 1.25°$, 85 levels to 85 km | 1 | 30 | | Bi et al. (submitted) |
| CanESM5 | $2.81° \times 2.81°$, 49 levels to 1 hPa | 1 | 50 | all | Swart et al. (2019) |
| CESM2 | $1.25° \times 0.9°$, 32 levels to 2.25 hPa | 1 | 30 | all | Danabasoglu et al. (2020) |
| CNRM-CM6-1 | $1.4° \times 1.4°$, 91 levels to 0.01 hPa | 5 | 30 | $CO_2$, ghg, aer, anthro | Voldoire et al. (2019) |
| CNRM-ESM2-1 | $1.4° \times 1.4°$, 91 levels to 0.01 hPa | 15 | 30 | all | Séférian et al. (2019) |
| EC-Earth3 | $0.7° \times 0.7°$, 91 levels to 0.01 hPa | 1 | 30 | | Wyser et al. (2019) |
| GFDL-CM4 | $1.25° \times 1°$, 33 levels to 1 hPa | 1 | 30 | all | Held et al. (2019) |
| GFDL-ESM4 | $1.25° \times 1°$, 49 levels to 1 hPa | 1 | 30 | | Dunne et al. (in prep.) |
| GISS-E2-1-G[1] | $2.5° \times 2°$, 40 levels to 0.1 hPa | 5 | 31/41[2] | | Kelley et al. (submitted) |
| HadGEM3-GC31-LL | $1.875° \times 1.25°$, 85 levels to 85 km | 1 | 30 | all | Williams et al. (2018) |
| IPSL-CM6A-LR | $2.5° \times 1.27°$, 79 levels to 80 km | 10 | 30 | all | Boucher et al. (submitted) |
| MIROC6 | $1.4° \times 1.4°$, 81 levels up to 0.004 hPa | 1 | 30 | aer | Tatebe et al. (2019) |
| MPI-ESM1-2-LR | $1.875° \times 1.875°$, 47 levels up to 0.01 hPa | 1 | 31 | | Mauritsen et al. (2019) |
| MRI-ESM2-0 | $1.125° \times 1.125°$, 80 levels to 0.01 hPa | 1 | 30 | all | Yukimoto et al. (2019) |
| NorESM2-LM | $2.5° \times 1.875°$, 32 levels to 3 hPa | 1 | 30 | | Seland et al. (2020) Kirkevåg et al. (2018) |
| NorESM2-MM | $1.25° \times 0.9375°$, 32 levels to 3 hPa | 1 | 30 | | Seland et al. (2020) |
| UKESM1-0-LL | $1.875° \times 1.25°$, 85 levels to 85 km | 3 | 45 | all | Sellar et al. (2019) |

1. GISS-E2-1-G produced two physics variants for piClim-control and piClim-aer; physics_version=1 (p1) includes aerosol and ozone specified by pre-computed transient fields and physics_version=3 (p3) includes aerosol-cloud interactions. Both physics versions are analysed in this paper and treated as separate models.

3. 41 years for r1i1p3f1

needed to diagnose ERF. Two advantages of this is that it reduces the computational burden for modelling centres, and can also be used to diagnose forcings of the order of 0.1 W m$^{-2}$ (Forster et al., 2016). For this reason, the climatological-SST method is implemented to derive forcing in RFMIP, and ERF in this paper (without qualifier) is taken to mean this.

The climatological-SST method of deriving ERF includes the TOA flux changes resulting from land-surface warming or cooling as part of the ERF. Conceptually, any land-surface temperature change as a response to forcing should be excluded in the same way that SST changes are (Shine et al., 2003; Hansen et al., 2005; Vial et al., 2013), but prescribing land surface temperatures is difficult in GCMs and this has not been performed in RFMIP. In essence, the goal is to completely isolate the forcing from any surface temperature change ($\Delta T$) or feedbacks ($\lambda$) in eq. (1). We test several methods to correct for adjustments to attempt to isolate forcing at $\Delta T = 0$ (also performed in Richardson et al. (2019); Tang et al. (2019)):

– *Effective radiative forcing (ERF)* is reserved to mean the TOA flux difference between a perturbed and control simulation, with climatological SSTs and sea ice distributions and no correction for land surface temperature change, as in Hansen et al. (2005); Myhre et al. (2013); Forster et al. (2016); Smith et al. (2018b).

– *Effective radiative forcing using a Gregory regression (ERF_reg)* is calculated from each model's CMIP abrupt4xCO2 experiment by regressing the annual temperature anomaly compared to the same model's pre-industrial control (piCon-

trol) against the annual TOA energy imbalance anomaly $\Delta N$ (eq. (1)) and finding the intercept at $\Delta T = 0$, as in Gregory et al. (2004). This is done for the first 20 years of model output to avoid the changing value of $\lambda$ over time present in many models (Armour, 2017); using the full 150 years tends to underestimate the forcing (denoted ERF_reg150; table S1). It is only possible to determine ERF_reg for $4 \times CO_2$ as coupled abrupt forcing experiments are not performed for other forcing agents as part of CMIP6.

– *Stratospherically adjusted radiative forcing (RF):* All tropospheric and surface adjustments, calculated using radiative kernels (section 4) are subtracted from the ERF, leaving just the stratospheric temperature adjustment to the IRF. The RF is included for historical comparison, although it is usually calculated using an offline method such as fixed dynamical heating (Forster and Shine, 1997). It should be noted that the stratospheric adjustment is included in all definitions of ERF.

– *Land-surface corrected effective radiative forcing (ERF_ts):* Land surface temperature change adjustment is subtracted from the climatological-SST ERF using the surface temperature radiative kernel.

– *Tropospherically corrected effective radiative forcing (ERF_trop)*: In addition to land-surface warming a proportion of tropospheric temperature and water vapour change is subtracted from the ERF using radiative kernels, by assuming a fixed lapse rate in the troposphere based on the land surface temperature change. The remaining tropospheric temperature

change when the constant lapse rate is subtracted is treated as the tropospheric temperature adjustment. The water vapour correction from the land surface warming is taken as the fraction of the adjustment from the constant lapse rate to the total tropospheric temperature adjustment. The surface albedo change is also removed, whereas no cloud adjustment is included justified by cloud adjustments to a large extent depending on heating/cooling in the troposphere (Smith et al., 2018b). This was known as ERF_kernel in Tang et al. (2019).

– *Feedback corrected effective radiative forcing (ERF_$\lambda$):* An amount corresponding to the global-average near-surface air temperature (GSAT) warming multiplied by the model's climate feedback parameter from its corresponding CMIP abrupt4xCO2 run is subtracted from the fixed-SST ERF. The same value of $\lambda$ from abrupt4xCO2 is applied to the GSAT change in all experiments. This method was first investigated by Hansen et al. (2005) and is known as ERF_fSST_$\Delta$Tland in Tang et al. (2019).

Table 2 shows the ERF diagnosed from each forcing and each model using the climatological-SST method, and fig. 1 shows the ERF, diagnosed IRF, and adjustments from each RFMIP Tier 1 experiment. Values for the different methods for calculating

**Table 2.** Effective radiative forcing from each Tier 1 time-slice RFMIP experiment for each model (W m$^{-2}$). Also shown is the $4\times CO_2$ ERF scaled to 2014 concentrations (as $1.4\times CO_2$) and the residual forcing (anthropogenic − WMGHGs − aerosol − land-use). WMGHGs = well-mixed greenhouse gases. Note that not all models performed all experiments.

| # | Model | $4\times CO_2$ | $1.4\times CO_2$ | WMGHGs | aerosols | land-use | anthropogenic | residual |
|---|---|---|---|---|---|---|---|---|
| 1 | ACCESS-CM2 | 7.95 | 1.80 | 3.04 | −1.09 | | 1.90 | |
| 2 | CanESM5 | 7.61 | 1.72 | 2.87 | −0.85 | −0.08 | 2.37 | 0.43 |
| 3 | CESM2 | 8.91 | 2.02 | 3.03 | −1.37 | −0.04 | 2.05 | 0.43 |
| 4 | CNRM-CM6-1 | 8.00 | 1.81 | 2.74 | −1.15 | | 1.61 | |
| 5 | CNRM-ESM2-1 | 7.93 | 1.80 | 2.51 | −0.74 | −0.07 | 1.66 | −0.04 |
| 6 | EC-Earth3 | 8.09 | 1.83 | 2.75 | −0.80 | −0.13 | 2.09 | 0.28 |
| 7 | GFDL-CM4 | 8.24 | 1.87 | 3.13 | −0.73 | −0.33 | 2.34 | 0.27 |
| 8 | GFDL-ESM4 | 7.74 | 1.75 | 3.23 | −0.70 | −0.28 | 2.17 | −0.08 |
| 9 | GISS-E2-1-G p1 | 7.35 | 1.67 | 2.89 | −1.32 | −0.00 | 1.93 | 0.35 |
| 10 | GISS-E2-1-G p3 | | | | −0.93 | | | |
| 11 | HadGEM3-GC31-LL | 8.09 | 1.83 | 3.11 | −1.10 | −0.11 | 1.81 | −0.08 |
| 12 | IPSL-CM6A-LR | 8.00 | 1.81 | 2.82 | −0.63 | −0.05 | 2.32 | 0.18 |
| 13 | MIROC6 | 7.32 | 1.66 | 2.69 | −1.04 | −0.03 | 1.80 | 0.17 |
| 14 | MPI-ESM1-2-LR | 8.35 | 1.89 | 2.69 | | −0.10 | 2.13 | |
| 15 | MRI-ESM2-0 | 7.65 | 1.73 | 3.03 | −1.21 | −0.17 | 1.95 | 0.29 |
| 16 | NorESM2-LM | 8.15 | 1.85 | 2.80 | −1.21 | 0.26 | 2.06 | 0.20 |
| 17 | NorESM2-MM | 8.38 | 1.90 | | −1.26 | | | |
| 18 | UKESM1-0-LL | 7.94 | 1.80 | 2.95 | −1.11 | −0.18 | 1.79 | 0.12 |
| | Mean | 7.98 | 1.81 | 2.89 | −1.01 | −0.09 | 2.00 | 0.20 |
| | Standard dev. | 0.38 | 0.09 | 0.19 | 0.23 | 0.13 | 0.23 | 0.17 |

forcing are given in Tables S1–S5. Instantaneous forcing (IRF) is calculated as the difference of the ERF and the sum of adjustments, with exception being land-use forcing where IRF is calculated directly from the surface albedo kernel. In keeping with the definitions of ERF and adjustments, IRF is defined at the TOA in this study. Adjustment calculations are explained in detail in section 4.

For ease of comparison we show $1.4\times CO_2$ instead of $4\times CO_2$, with the scaling to present-day concentrations assumed to apply to ERF, IRF and all adjustments proportionally. Figure 1 also shows the ERF_reg (for $4\times CO_2$), ERF_ts, ERF_$\lambda$ and RF. In general, the methods that correct for land surface temperature change (ERF_ts, ERF_trop and ERF_$\lambda$) result in forcings that are slightly stronger than non-corrected ERF, although differences between these methods are comparable to the magnitude of internal year-to-year variability and small compared to the contribution of adjustments. For $CO_2$, ERF_reg results in a similar

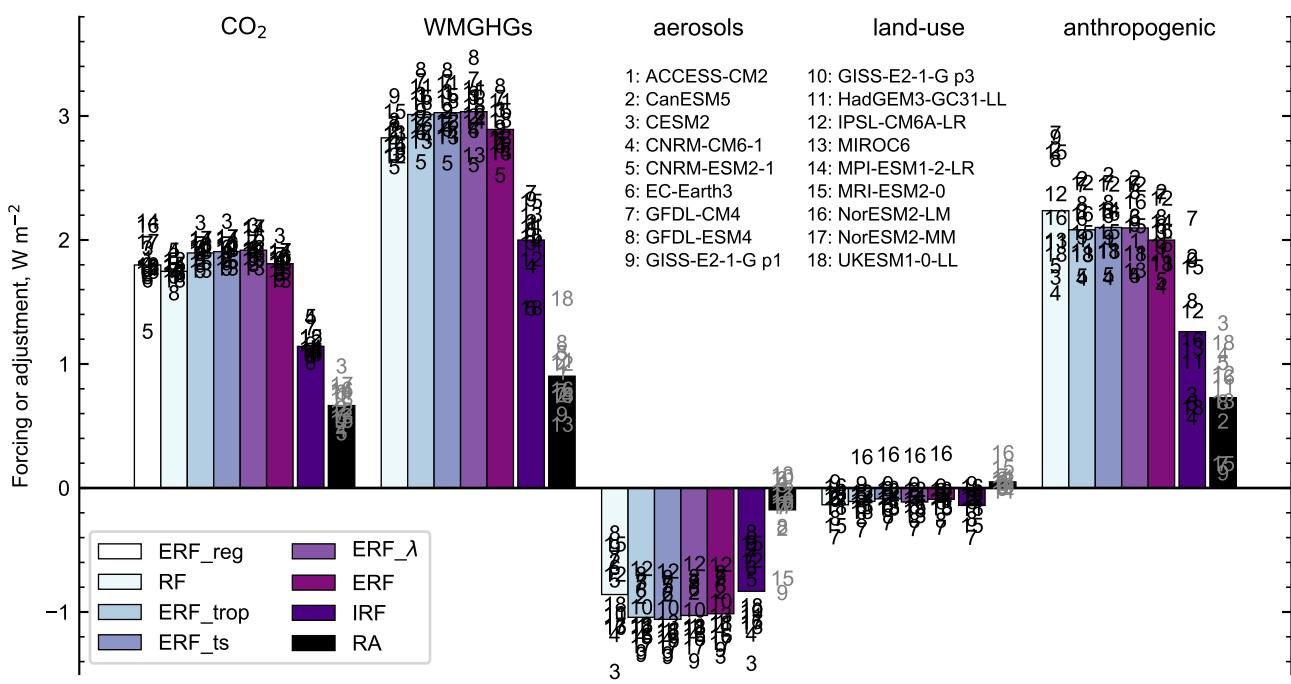

**Figure 1.** Comparison of radiative forcing (RF, which by definition includes stratospheric temperature adjustment), effective radiative forcing with tropospheric correction (ERF_trop), effective radiative forcing with land-surface kernel correction (ERF_ts), feedback-corrected ERF (ERF_$\lambda$), and fixed-SST ERF. For $CO_2$ forcing, ERF from a Gregory regression (ERF_reg) from each model's corresponding abrupt4xCO2 CMIP simulation is also given. The ERF is compared with the IRF and adjustments (RA) for each of the present-day RFMIP-ERF time slice experiments ($1.4\times CO_2$ is shown instead of $4\times CO_2$ for better comparison with other forcing agents). Individual models are numbered.

mean estimate of ERF to the fixed-SST method. Excluding CNRM-ESM2-1 for reasons described in the next section, the $4\times CO_2$ ERF_reg is 8.09 W m$^{-2}$ compared to 7.99 W m$^{-2}$ for ERF.

## 4   Forcing adjustments

### 4.1   Non-cloud adjustments

Adjustments to the radiative forcing describe flux changes resulting from changing atmospheric or surface state, in response to a forcing, but unrelated to the change in globally-averaged surface temperature (thus decoupling them from climate feedbacks, Myhre et al. (2013); Sherwood et al. (2015)). Adjustments to non-cloud changes in this study are calculated using radiative kernels (Shell et al., 2008; Soden et al., 2008; Block and Mauritsen, 2013; Huang, 2013; Chung and Soden, 2015; Vial et al., 2013; Smith et al., 2018b; Pendergrass et al., 2018). The difference in an atmospheric state variable $x$ (air temperature, surface

temperature, specific humidity or surface albedo) between a forcing perturbation (pert) and piClim-control (base) is multiplied

by the kernel $K_x$ to derive the adjustment $A_x$:

$$A_x = K_x(x_{\text{pert}} - x_{\text{base}}) \tag{2}$$

The radiative kernel describes the change in TOA fluxes for a unit change in state for $x \in \{T, T_s, q, \alpha\}$ where $T$ is atmospheric air temperature, $T_s$ is surface temperature, $q$ is water vapour and $\alpha$ is surface albedo. $K_T$ and $K_q$ are four-dimensional (month, pressure level, latitude, longitude) and $K_{T_s}$ and $K_\alpha$ are three dimensional (month, latitude, longitude). Kernels are produced for both longwave and shortwave radiation changes. Typical unit changes are 1 K for temperature, the change in specific humidity that maintains constant relative humidity for a temperature increase of 1 K for water vapour, and 1% additive for surface albedo. For the division of temperature into stratospheric and tropospheric components, the WMO definition of the lapse-rate tropopause is used from each model's piClim-control run, using geopotential height as an approximation of geometric height on model pressure levels.

The water vapour kernel describes the change in TOA flux for a perturbation that maintains relative humidity for a temperature increase of 1 K, the effect being that specific humidity increases. The assumption therefore is that relative humidity is approximately constant between perturbation and control runs, which is found to be true in coupled experiments (Held and Soden, 2000; Held and Shell, 2012). Note that the difference in states is taken for the logarithm of water vapour concentration in eq. (2). More details on the application of the kernel method can be found in Smith et al. (2018b, Supplementary Material).

In this paper we use radiative kernels derived from the atmospheric component of the HadGEM3-GC31-LL model (HadGEM3-GA7.1), interpolated to the 19 standard CMIP6 pressure levels (Smith et al., 2020). With the exception of stratospheric temperature adjustments to greenhouse-gas forcing, structural differences introduced by using different kernels are well within 0.1 W m$^{-2}$ (Soden et al., 2008; Smith et al., 2018b), and the HadGEM3-GA7.1 kernel is representative of the population of radiative kernels commonly used in the literature for tropospheric and surface adjustments (fig. S1); we use this particular kernel for its improved stratospheric resolution as outlined in Smith et al. (2020).

Stratospheric adjustments to greenhouse-gas driven experiments are expected to equilibrate within a few model months (Sherwood et al., 2015). We find that the time to reach equilibrium varies between models for a 4×CO$_2$ forcing. Figure 2 shows the time taken for the stratospheric temperature adjustment, and hence stratospheric temperatures, to adjust to a 4×CO$_2$ forcing. In CNRM-ESM2-1, concentrations of CO$_2$ are relaxed towards the 4× pre-industrial level below 560 hPa, and allowed to propagate throughout the atmosphere, therefore taking around 15 years to reach an approximate uniform atmospheric concentration. A similar specification is implemented in the abrupt-4xCO2 run of CNRM-ESM2-1, causing ERF_reg to be biased low (fig. 1). This highlights one advantage of the fixed-SST based methods over the Gregory regression, as these "spin-up" years can simply be discarded with a fixed-SST measure of ERF. The CO$_2$ treatment in CNRM-ESM2-1 is in contrast to the physical climate model from the same group (CNRM-CM6-1). However, even in some physical models, we find that the time to reach equilibrium varies between models and may be up to 10 years (e.g. in IPSL-CM6A-LR; fig. 2). For this reason, we discard the first few years of model output where the stratosphere is still adjusting to a forcing for the 4×CO$_2$, well-mixed greenhouse gas (WMGHG) and anthropogenic forcing experiments (table 1). We find this issue is not present in the aerosol or land-use experiments. It is important to emphasise that our stratospheric adjustment is calculated in a different way to the usual

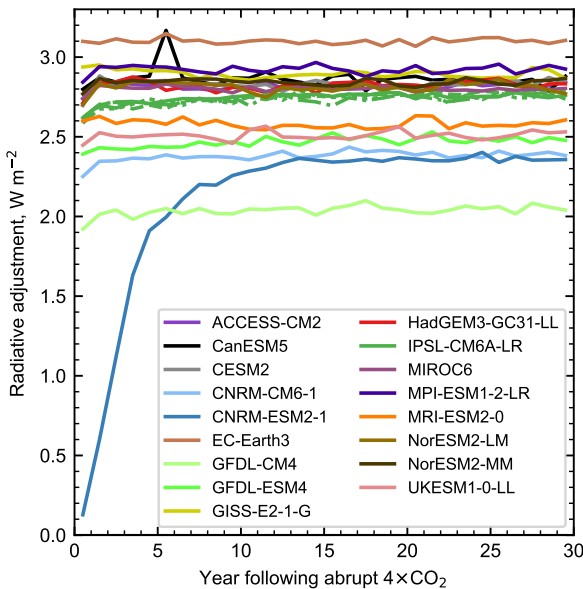

**Figure 2.** Transient response of the stratospheric temperature adjustment to a $4\times CO_2$ forcing. The small spike in year 6 in CanESM5 is due to an unseasonably low tropical tropopause in July of year 6, resulting in much of the temperature adjustment at the 100 hPa level to be counted in the stratosphere.

RF method which uses an offline radiative transfer method. It may therefore be the case that differences are due to a change in tropopause height in greenhouse-gas-driven experiments (Santer et al., 2003).

## 4.2 Cloud adjustments

The radiative effect of clouds depends on their coverage (both within layer and total), ice water content, liquid water content, droplet effective radius and ice particle habit. Cloud properties vary extensively from model to model, and unlike pressure level diagnostics of temperature and humidity, cloud diagnostics are not output on 19 standard pressure levels in CMIP. A number of different approaches have therefore been used to estimate cloud adjustments, depending on availability of diagnostics and model specific setup, and we can exploit methods originally designed for cloud feedback calculations for calculating adjustments. Where cloud adjustments can be calculated with more than one method, we take the mean of each available method. In some models and experiments, cloud adjustments cannot be calculated and no estimate is made.

### 4.2.1 ISCCP simulator kernel

The ISCCP simulator (Klein and Jakob, 1999; Webb et al., 2001) provides a joint $7\times 7$ histogram of cloud visble-wavelength optical depth ($\tau$) and cloud top pressure (CTP). These outputs can be multiplied by the ISCCP simulator kernel (Zelinka et al.,

2012) to estimate the impact of cloud changes on top-of-atmosphere fluxes. Ten models included ISCCP simulator diagnostics within their RFMIP output (table 1).

The ISCCP simulator kernel reports all flux changes resulting from clouds. For $CO_2$, WMGHG and land-use forcings, it is assumed that cloud droplet effective radius does not change (except for the land-use experiment in NorESM2-LM as discussed in section 5.4, but this model did not include ISCCP simulator diagnostics), and therefore in these experiments the SW flux changes from the ISCCP simulator kernel are treated as the cloud adjustment. For aerosol and total anthropogenic forcing this is usually not the case as most models include aerosol-radiation interactions (significant in the SW), with ice particle behaviour also changing in the MRI-ESM-2.0, MIROC6 and CESM2 models which affects LW fluxes. NorESM2-LM also includes the effects of mineral dust and BC on heterogeneous ice nucleation (Kirkevåg et al., 2018). Following Boucher et al. (2013) we treat the cloud-albedo response to aerosols as part of the IRF, and the ISCCP simulator kernel is unable to separate this effect from any adjustment. We assume that any LW effect from aerosol-cloud interactions is small except in those models that include aerosol effects on ice clouds.

### 4.2.2 Approximate partial radiative perturbation with liquid water path adjustment

The approximate partial radiative perturbation (APRP; section 5.3.3) method uses standard climate model diagnostics to estimate the components of SW ERF attributed to cloud fraction change, and all-sky and clear-sky scattering and absorption. With no changes in aerosol forcing, the changes in cloud absorption, cloud scattering and cloud amount calculated from APRP can be taken to be the SW cloud adjustment. We use this estimate for $CO_2$, WMGHG and land-use forcing.

For aerosol forcing, the effect of cloud amount changes calculated by APRP ($A_{\mathrm{CLT}}$) is an adjustment, but the cloud scattering is a combination of radiative forcing due to aerosol-cloud interactions (RFaci), treated as part of the IRF, and adjustments due to cloud liquid water path (LWP) changes ($A_{\mathrm{LWP}}$; Bellouin et al., 2020b). For the LWP adjustment we use a relationship obtained in Gryspeerdt et al. (2019) in which LWP adjustment (W m$^{-2}$) scales linearly with vertically integrated in-cloud liquid water path (kg m$^{-2}$):

$$A_{\mathrm{LWP}} = -\frac{1000}{37.6} \left( \frac{\mathrm{clwvi_{pert}} - \mathrm{clivi_{pert}}}{\mathrm{clt_{pert}}/100} - \frac{\mathrm{clwvi_{base}} - \mathrm{clivi_{base}}}{\mathrm{clt_{base}}/100} \right). \tag{3}$$

In eq. (3), clwvi, clivi and clt are the CMIP6 variable labels for total cloud water path, ice water path and total cloud fraction in percent. We then isolate the RFaci as

$$\mathrm{RFaci} = \mathrm{ERFaci} - A_{\mathrm{LWP}} - A_{\mathrm{CLT}}. \tag{4}$$

with ERFaci, the effective radiative forcing due to aerosol-cloud interactions, calculated from APRP (section 5.3.3).

For anthropogenic total forcing, the RFaci calculated in eq. (4) from the aerosol forcing experiment is subtracted from the total derived cloud change under APRP, which includes contributions from greenhouse gases and land use as well as RFaci. For models not including ice cloud nucleation, the LW cloud adjustment for aerosols is estimated from the change in cloud radiative effect (CRE; difference between all-sky and clear-sky fluxes). For other experiments this results in a biased estimate of cloud adjustment due to masking of LW adjustments.

### 4.2.3 Offline monthly mean partial radiative perturbation

A direct estimate of cloud radiative effect can be obtained by substituting model cloud fields into an offline radiative transfer model. We perform these offline calculations using the SOCRATES radiative transfer code (Edwards and Slingo, 1996). This is produced by substituting fields of 3D cloud fraction, cloud water content and cloud ice content from each model and experiment into a climatology for the year 2014 provided by ERA5 (Copernicus Climate Change Service, 2017). Taking the cloud fields in each experiment minus those from the control gives $A_{\mathrm{LWP}} + A_{\mathrm{CLT}}$ in each model. As only monthly mean diagnostics are available from models in general, we only attempt this in the LW which is assumed to be less biased than the SW (Mülmenstädt et al., 2019; Bellouin et al., 2020a). The monthly mean cloud fraction, ice water content and liquid water content variables in all experiments are scaled by a model-dependent factor that ranges between 0.68 and 1.5 to ensure that TOA LW outgoing flux is approximately 240.2 W m$^{-2}$ in the control experiment, in line with TOA observations (Loeb et al., 2018).

### 4.2.4 Kernel masking

In the land-use experiment, IRF is directly estimated from the surface albedo kernel such that IRF = $A_\alpha$. As there are no other unknowns in the kernel decomposition, cloud adjustments can be calculated using the difference between all-sky and clear-sky fluxes (Soden et al., 2008), such that

$$A_c = (\mathrm{ERF} - \mathrm{ERF}^{\mathrm{clr}}) - (A_\alpha - A_\alpha^{\mathrm{clr}}) - \sum_{i \in \{\mathrm{T}, \mathrm{T_s}, \mathrm{q}\}} (A_i - A_i^{\mathrm{clr}}) \tag{5}$$

where the clr superscript in eq. (5) refers to fluxes calculated with clear-sky radiative kernels.

## 5 Multi-model results

Figure 3 shows the contribution to the total adjustment in each experiment from land surface temperature, tropospheric temperature, stratospheric temperature, water vapour, surface albedo and clouds. No corrections for tropospheric or land surface warming as discussed in section 3 have been performed for these results.

Figure 4 shows the effect on TOA radiative flux arising from cloud responses from the ISCCP simulator for each experiment from models that provided these diagnostics (table 1). In this figure, histogram boxes not marked with a cross are where 75% or more of the models agree on the sign of the cloud fraction or radiative flux change, following Zelinka et al. (2012).

### 5.1 Carbon dioxide

The multi-model mean ERF from a quadrupling of $CO_2$ is 7.98 W m$^{-2}$ ($\pm$ 0.38 W m$^{-2}$; all ranges given as one standard deviation). A point of comparison for ERF is Etminan et al. (2016), who computed estimates of and parametric fits for radiative forcing accounting for masking by clouds and stratospheric temperature equilibration, using a tropical and mid-latitude profile to represent the global mean. The implied ERF from RFMIP models for 2×$CO_2$ is 3.81 ($\pm$ 0.18) W m$^{-2}$ when scaling down

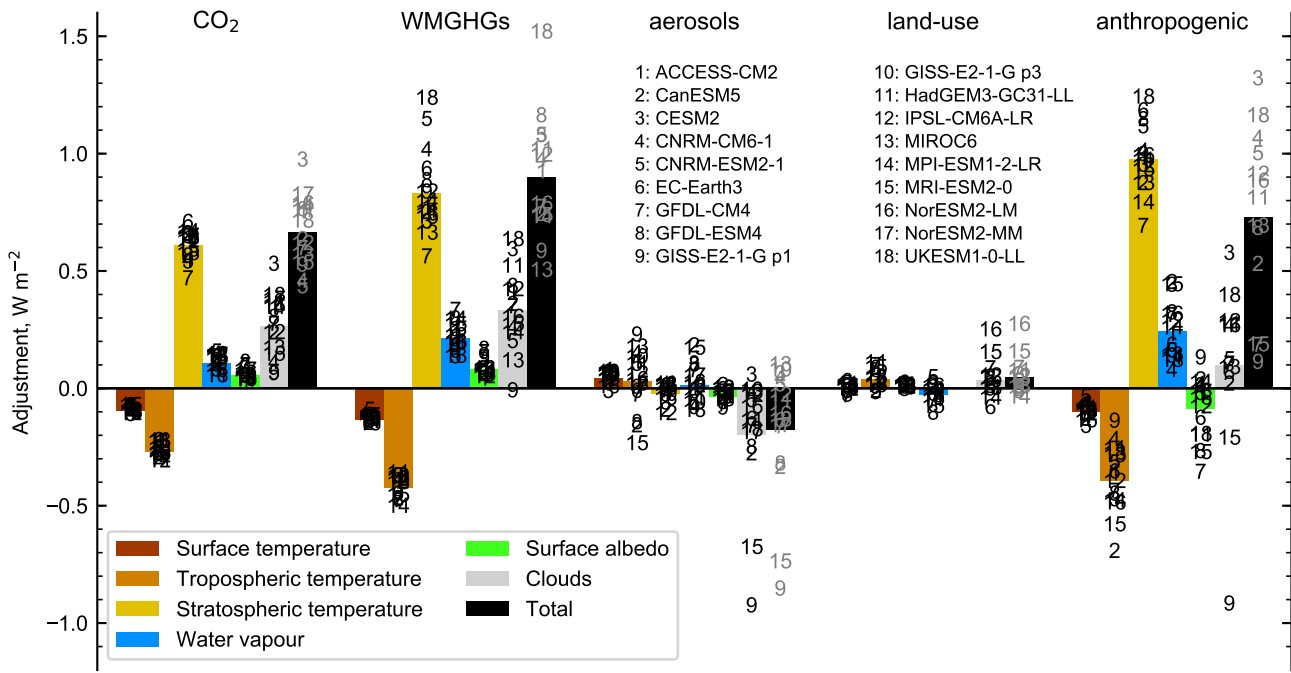

**Figure 3.** Adjustments broken down by mechanism in each of the present-day RFMIP-ERF time slice experiments. Black/grey numbers indicate individual models, coloured bars indicate the multi-model mean.

the $4\times CO_2$ results using the Etminan et al. (2016) formula, comparable to a radiative forcing of 3.80 W m$^{-2}$ for a doubling of $CO_2$ in Etminan et al. (2016). Both estimates are slightly higher than the best estimate of 3.71 W m$^{-2}$ from the IPCC's Fifth Assessment Report (AR5; Myhre et al. (2013)). Scaling down the $4\times CO_2$ forcing using Etminan et al. (2016), our derived multi-model mean for $1.4\times CO_2$ is 1.81 ($\pm$ 0.09) W m$^{-2}$. As shown in fig. 1 and discussed in section 5.1, ERF is approximately equal to RF for $CO_2$, and we apply the Etminan formula to ERF.

The $4\times CO_2$ ERF from 17 CMIP6 models is larger, but not significantly so ($p$-value 0.13 using a Welch's $t$-test), than the $4\times CO_2$ ERF from 13 CMIP5 models of 7.53 ($\pm$0.89) W m$^{-2}$ (Kamae and Watanabe, 2012). In addition, CMIP6 models are notable for their smaller spread in $CO_2$ ERF than CMIP5 models (fig. 5). Zelinka et al. (2020) show that ERF_reg150 for $4\times CO_2$ also increases in CMIP6 compared to CMIP5, and attribute 20% of the increase in multi-model mean effective climate sensitivity (ECS) in CMIP6 to this. We note that a long standing problem in GCMs has been on the diversity in the forcing of $CO_2$ (Soden et al., 2018), which may result both from model broadband radiation parameterisation error in the IRF component (Pincus et al., 2015) and differences in base state climatology between models. The reduction in spread of $CO_2$ forcing in CMIP6 may be indicative that model radiation parameterisations are improving, for example as documented in HadGEM3-GC31-LL and UKESM1-0-LL (Andrews et al., 2019), but could also be from a convergence in model base states, including clouds.

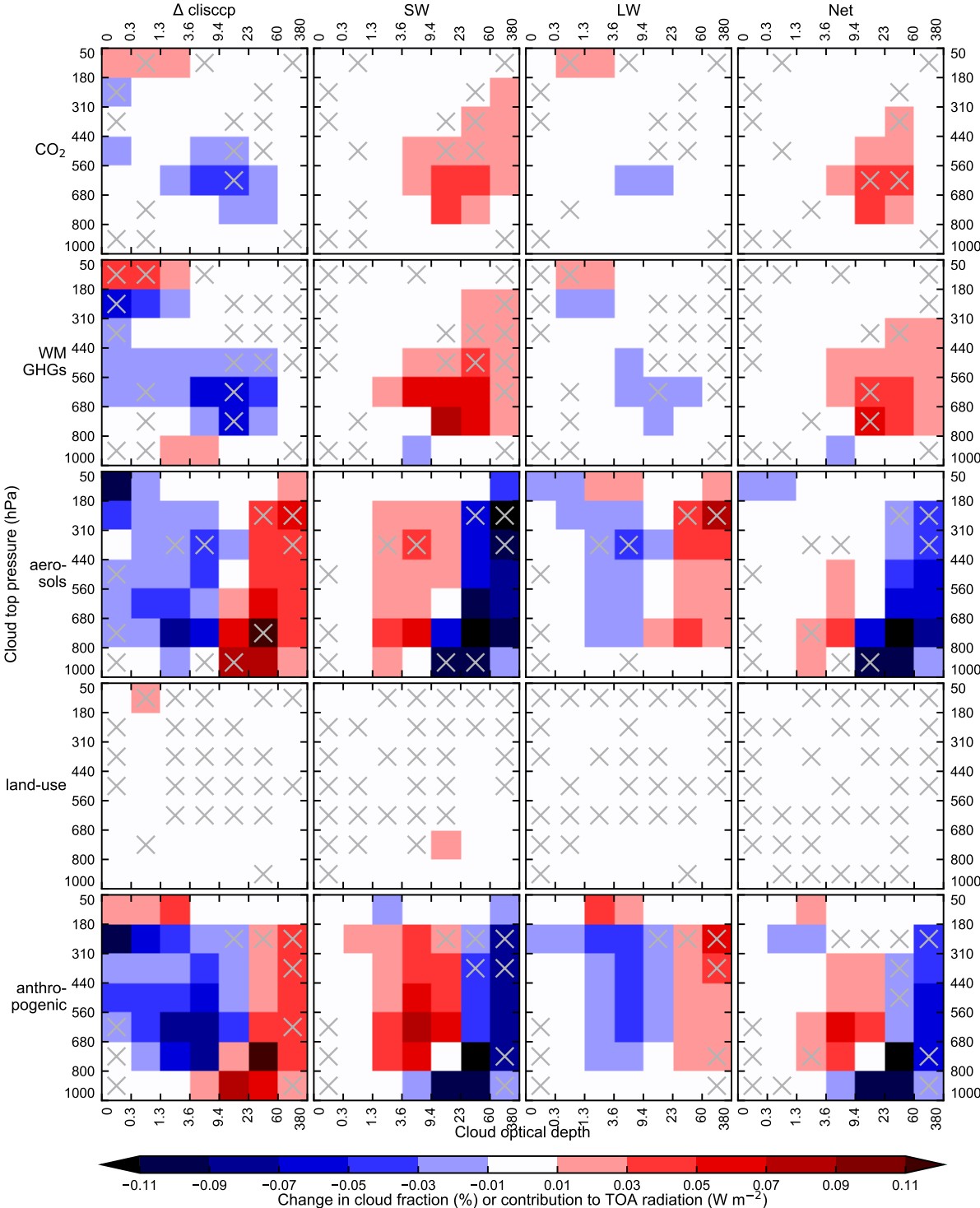

**Figure 4.** Global mean change in ISCCP-simulated cloud fraction in CTP-$\tau$ space (first column) and consequential changes in SW (second column), LW (third column) and net (fourth column) radiation when convoluted with the ISCCP cloud kernel. Grey crosses show where less than 75% of models agree on sign. Figure shows the multi-model mean cloud fraction and radiative effect. For $1.4\times CO_2$ the change in cloud fraction, as well as the radiative fluxes, are scaled down from the $4\times CO_2$ experiment using Etminan et al. (2016).

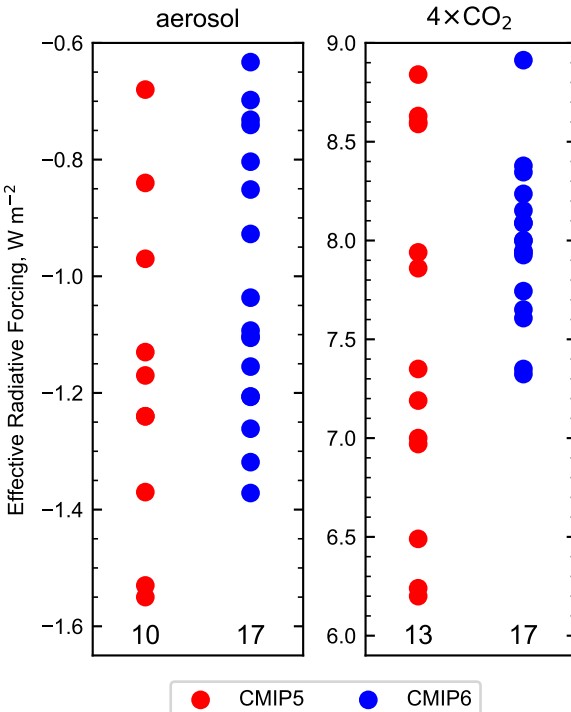

**Figure 5.** Aerosol and $4\times CO_2$ effective radiative forcing from CMIP5 sstClim4xCO2 and sstClimAerosol experiments (Kamae and Watanabe, 2012; Zelinka et al., 2014) and CMIP6 RFMIP experiments. Numbers at the bottom of each plot give the number of participating models.

The breakdown of ERF into adjustments is shown in table 3 with the corresponding $4\times CO_2$ values in table S6. Stratospheric temperature adjustment dominates for $CO_2$-driven simulations, which is well-known (Smith et al., 2018b; Myhre et al., 2013).

Tropospheric adjustments approximately sum to zero, such that the overall adjustment approximately equals the stratospheric adjustment, and RF is a good approximation to ERF (Smith et al., 2018b). Nevertheless, individual tropsopheric adjustments are non-zero and significant. A warming land surface and troposphere leads to a negative adjustment (more outgoing LW radiation to space) that is partially offset by increased tropospheric water vapour (analogous to the water vapour feedback). Cloud adjustments are overall positive, dominated by a reduction in mid-troposphere clouds driven by tropospheric warming,

leading to a positive SW radiative effect (fig. 4). The LW effect is small in comparison, so that the SW effect dominates the net cloud adjustment.

The spatial pattern of adjustments is shown in fig. 6. In figs. 6 to 8, 12 and 13, cloud changes are only shown from the ISCCP simulator kernels in subfigures (g–i) and are not the means of all participating models, whereas ERF and non-cloud adjustments in (a–f) are multi-model means. Hatched areas are defined where less than 75% of models agree on the sign of the

295 change. Stratospheric cooling is spatially uniform and results in positive adjustment of $+0.61$ W m$^{-2}$, i.e. around one third

**Table 3.** ERF, IRF and adjustments (W m$^{-2}$) by component from 1.4×CO$_2$, scaled down from the 4×CO$_2$ RFMIP experiment. ts=surface temperature, ta_tr=tropospheric temperature, ta_st=stratospheric temperature, hus=water vapour, albedo=surface albedo, cl=clouds.

| # | Model | ERF | IRF | Adj. | ts | ta_tr | ta_st | hus | albedo | cl |
|---|-------|-----|-----|------|-----|-------|-------|-----|--------|-----|
| 1 | ACCESS-CM2 | 1.80 | 1.05 | 0.75 | −0.09 | −0.23 | 0.64 | 0.07 | 0.02 | 0.34 |
| 2 | CanESM5 | 1.72 | 1.09 | 0.63 | −0.10 | −0.30 | 0.65 | 0.10 | 0.05 | 0.23 |
| 3 | CESM2 | 2.02 | 1.05 | 0.97 | −0.12 | −0.29 | 0.64 | 0.11 | 0.09 | 0.52 |
| 4 | CNRM-CM6-1 | 1.81 | 1.36 | 0.45 | −0.10 | −0.29 | 0.54 | 0.14 | 0.05 | 0.11 |
| 5 | CNRM-ESM2-1 | 1.80 | 1.37 | 0.43 | −0.08 | −0.28 | 0.53 | 0.15 | 0.05 | 0.06 |
| 6 | EC-Earth3 | 1.83 | | | −0.09 | −0.27 | 0.70 | 0.11 | 0.05 | |
| 7 | GFDL-CM4 | 1.87 | 1.28 | 0.59 | −0.09 | −0.28 | 0.46 | 0.13 | 0.09 | 0.27 |
| 8 | GFDL-ESM4 | 1.75 | 1.00 | 0.76 | −0.08 | −0.26 | 0.56 | 0.13 | 0.11 | 0.30 |
| 9 | GISS-E2-1-G p1 | 1.67 | 1.14 | 0.52 | −0.09 | −0.23 | 0.65 | 0.07 | 0.05 | 0.06 |
| 11 | HadGEM3-GC31-LL | 1.83 | 1.08 | 0.75 | −0.11 | −0.23 | 0.64 | 0.05 | 0.03 | 0.36 |
| 12 | IPSL-CM6A-LR | 1.81 | 1.20 | 0.61 | −0.11 | −0.31 | 0.62 | 0.14 | 0.04 | 0.23 |
| 13 | MIROC6 | 1.66 | 1.09 | 0.57 | −0.10 | −0.26 | 0.63 | 0.07 | 0.05 | 0.17 |
| 14 | MPI-ESM1-2-LR | 1.89 | 1.12 | 0.77 | −0.11 | −0.31 | 0.66 | 0.14 | 0.05 | 0.34 |
| 15 | MRI-ESM2-0 | 1.73 | 1.20 | 0.53 | −0.08 | −0.28 | 0.58 | 0.13 | 0.04 | 0.14 |
| 16 | NorESM2-LM | 1.85 | 1.07 | 0.78 | −0.11 | −0.28 | 0.64 | 0.12 | 0.06 | 0.35 |
| 17 | NorESM2-MM | 1.90 | 1.08 | 0.82 | −0.11 | −0.30 | 0.64 | 0.14 | 0.07 | 0.38 |
| 18 | UKESM1-0-LL | 1.80 | 1.09 | 0.71 | −0.11 | −0.23 | 0.57 | 0.05 | 0.03 | 0.39 |
| | Mean | 1.81 | 1.14 | 0.66 | −0.10 | −0.27 | 0.61 | 0.11 | 0.05 | 0.27 |
| | St. dev. | 0.09 | 0.11 | 0.14 | 0.01 | 0.03 | 0.06 | 0.03 | 0.02 | 0.13 |

of the total ERF. Tropospheric temperature adjustments are globally negative and robust. Cloud changes show several robust spatial patterns, including positive changes over Eurasia and North America land.

## 5.2 Well-mixed greenhouse gases

The ERF from all well-mixed greenhouse gases is evaluated to be 2.89 (±0.19 W m$^{-2}$) for 1850–2014, implying a contribution of 1.08 (±0.21) W m$^{-2}$ from non-CO$_2$ WMGHGs (uncertainties in quadrature and this definition excludes changes in ozone). Tier 1 of RFMIP does not contain additional granularity to break down non-CO$_2$ forcing by species, however dedicated experiments to derive ERF from methane, nitrous oxide and halocarbons separately are part of the protocol for the Aerosol and Chemistry Model Intercomparison Project (AerChemMIP; Thornhill et al., 2020; Collins et al., 2017).

There is also a substantial adjustment arising from WMGHG forcing, and again this is mostly driven by stratospheric cooling implied by the observation that ERF and RF are approximately equal. This confirms PDRMIP model behaviour for CO$_2$ and CH$_4$ forcing (Smith et al., 2018b), which found that tropospheric and land adjustments, while individually significant,

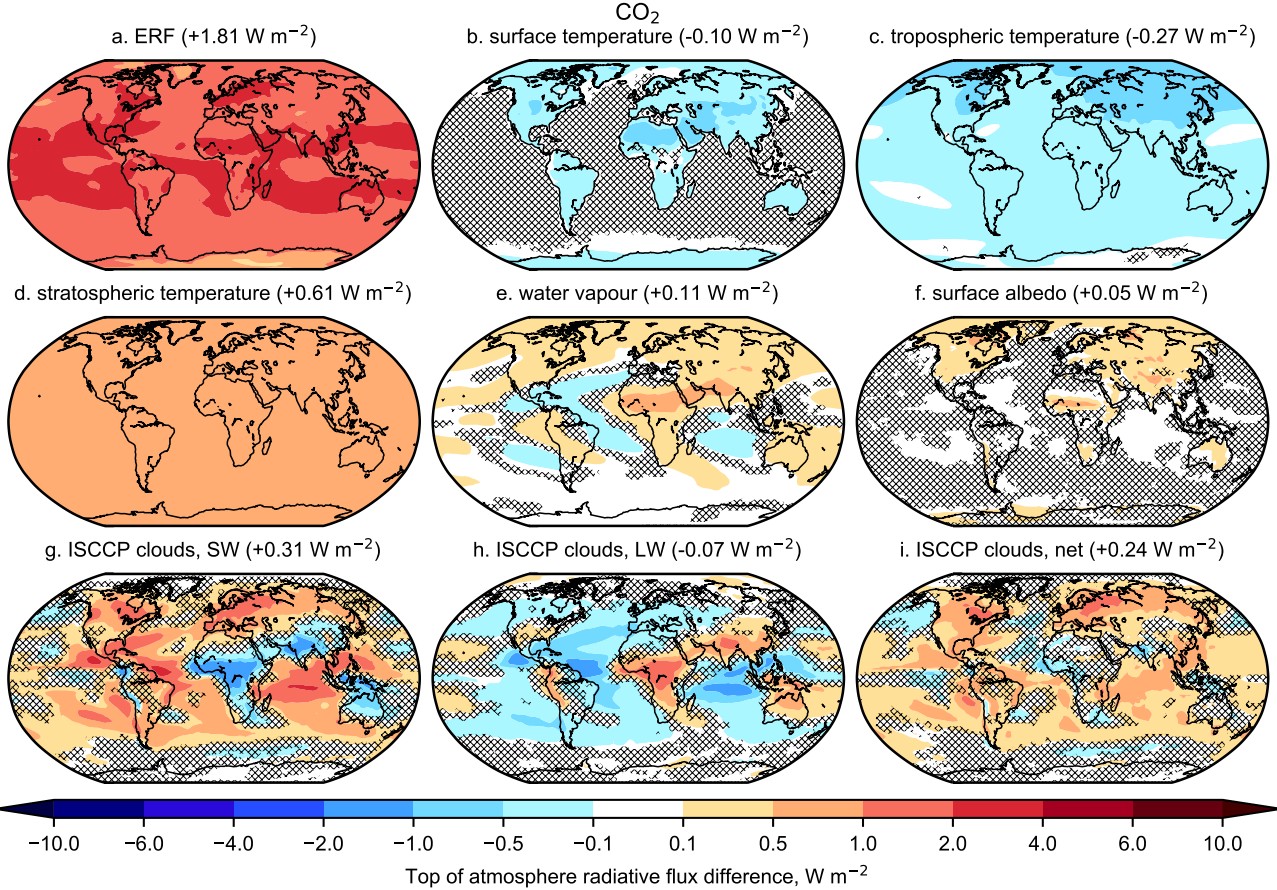

**Figure 6.** Multi-model mean spatial patterns of (a) effective radiative forcing, (b–f) adjustments and (g–i) cloud contributions to ERF for $1.4 \times CO_2$. Hatched regions are where less than 75% of models agree on the sign of the change.

approximately sum to zero leaving just the stratospheric temperature adjustment. Unlike in Smith et al. (2018b), who found that the stratospheric temperature adjustment to methane was approximately zero, we find a larger stratospheric temperature adjustment for WMGHGs compared to $CO_2$ implying a positive non-$CO_2$ WMGHG stratospheric adjustment, although this 310 cannot be attributed to individual gases.

The multi-model mean non-$CO_2$ WMGHG ERF of 1.08 W m$^{-2}$ is close to the 1850–2014 RF of 1.09 W m$^{-2}$ made up of $CH_4$ (0.55 W m$^{-2}$) plus $N_2O$ (0.17 W m$^{-2}$) from Etminan et al. (2016), plus halocarbons (0.37 W m$^{-2}$) using relationships from Myhre et al. (2013).

As for $CO_2$ only forcing, the total adjustment approximately equals the stratospheric temperature adjustment, implying that 315 tropospheric and surface adjustments approximately cancel (table 4) so that the spread in their sum is smaller than for each component individually. For the ISCCP-simulator cloud adjustments, a similar pattern can be seen from all WMGHGs to

**Table 4.** As for table 3 but for 1850–2014 well-mixed greenhouse gas forcing.

| # | Model | ERF | IRF | Adj. | ts | ta_tr | ta_st | hus | albedo | cl |
|---|---|---|---|---|---|---|---|---|---|---|
| 1 | ACCESS-CM2 | 3.04 | 2.12 | 0.92 | −0.13 | −0.40 | 0.77 | 0.24 | 0.04 | 0.41 |
| 2 | CanESM5 | 2.87 | 2.13 | 0.74 | −0.15 | −0.47 | 0.74 | 0.21 | 0.05 | 0.35 |
| 3 | CESM2 | 3.03 | 1.96 | 1.07 | −0.15 | −0.44 | 0.70 | 0.25 | 0.12 | 0.59 |
| 4 | CNRM-CM6-1 | 2.74 | 1.77 | 0.97 | −0.15 | −0.40 | 1.01 | 0.17 | 0.08 | 0.25 |
| 5 | CNRM-ESM2-1 | 2.51 | 1.43 | 1.07 | −0.10 | −0.38 | 1.14 | 0.13 | 0.08 | 0.20 |
| 6 | EC-Earth3 | 2.75 | | | −0.13 | −0.45 | 0.93 | 0.20 | 0.06 | |
| 7 | GFDL-CM4 | 3.13 | 2.37 | 0.77 | −0.13 | −0.48 | 0.56 | 0.33 | 0.15 | 0.34 |
| 8 | GFDL-ESM4 | 3.23 | 2.07 | 1.16 | −0.13 | −0.48 | 0.88 | 0.29 | 0.16 | 0.43 |
| 9 | GISS-E2-1-G p1 | 2.89 | 2.31 | 0.58 | −0.14 | −0.38 | 0.83 | 0.15 | 0.13 | −0.01 |
| 11 | HadGEM3-GC31-LL | 3.11 | 2.09 | 1.01 | −0.15 | −0.36 | 0.75 | 0.19 | 0.06 | 0.52 |
| 12 | IPSL-CM6A-LR | 2.82 | 1.83 | 0.99 | −0.13 | −0.39 | 0.83 | 0.22 | 0.05 | 0.42 |
| 13 | MIROC6 | 2.69 | 2.19 | 0.50 | −0.13 | −0.41 | 0.66 | 0.19 | 0.07 | 0.11 |
| 14 | MPI-ESM1-2-LR | 2.69 | 1.96 | 0.73 | −0.14 | −0.51 | 0.79 | 0.29 | 0.06 | 0.23 |
| 15 | MRI-ESM2-0 | 3.03 | 2.30 | 0.73 | −0.12 | −0.46 | 0.72 | 0.27 | 0.05 | 0.27 |
| 16 | NorESM2-LM | 2.80 | 2.02 | 0.78 | −0.14 | −0.39 | 0.74 | 0.19 | 0.08 | 0.30 |
| 18 | UKESM1-0-LL | 2.95 | 1.44 | 1.51 | −0.16 | −0.38 | 1.23 | 0.13 | 0.06 | 0.63 |
| | Mean | 2.89 | 2.00 | 0.90 | −0.14 | −0.42 | 0.83 | 0.22 | 0.08 | 0.34 |
| | St. dev. | 0.19 | 0.27 | 0.25 | 0.01 | 0.04 | 0.17 | 0.06 | 0.04 | 0.17 |

$CO_2$-only forcing, with a larger reduction in mid-troposphere cloud fraction leading to a greater positive SW adjustment which dominates the net adjustment.

The spread in ERF and stratospheric temperature adjustments is larger for WMGHG than for $CO_2$ forcing alone. One factor may be the inclusion or exclusion of stratospheric chemistry, which affects ozone formation. The effect can be seen by comparing Earth system (ESM) and physical models from the same group: the UKESM1-0-LL ESM (model 18) to the HadGEM3-GC31-LL physical model (model 11), and CNRM-ESM2-1 (model 5) to CNRM-CM6-1 (model 4). The physical models show ERFs around 0.2 W m$^{-2}$ greater than the ESMs, a greater IRF, and a smaller stratospheric temperature adjustment. Additionally, for UKESM1-0-LL, large and compensating ERFs from $CH_4$ (+0.93 W m$^{-2}$) and halocarbons (−0.33 W m$^{-2}$), resulting from interactive chemistry, bring the total WMGHG ERF closer to the no-chemistry ERFs total from HadGEM3-GC31-LL (O'Connor et al., 2020).

The spatial patterns are overall similar to the $CO_2$ experiment (fig. 7) with a larger magnitude.

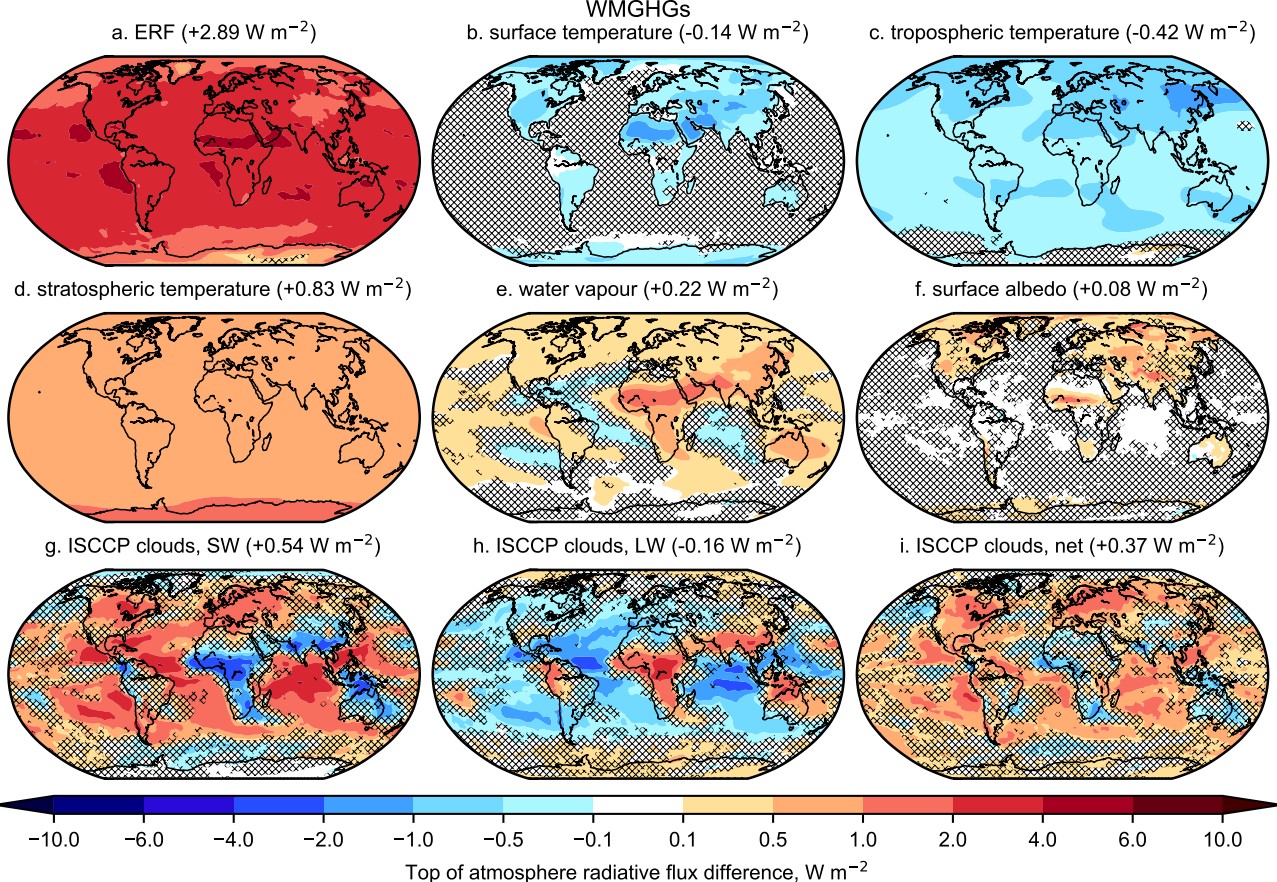

**Figure 7.** As fig. 6 but for present-day WMGHG forcing.

## 5.3 Aerosols

### 5.3.1 Forcing and adjustments

Present-day aerosol ERF is $-1.01$ ($\pm0.23$) W m$^{-2}$ from 17 models. The full range of aerosol ERF estimates for 2014 versus 1850 is $-0.63$ to $-1.37$ W m$^{-2}$. This is a narrower range of ERF than similar experiments performed with CMIP5 models for year 1850 and year 2000 forcings (Zelinka et al., 2014), particularly in relation to the lower (more negative) bound of aerosol forcing. Based on the 2000–1850 estimate of $-1.17$ ($\pm0.30$) W m$^{-2}$ from Zelinka et al. (2014), aerosol forcing in CMIP6 models is less negative than in CMIP5, but this difference again is not significant ($p$-value 0.15). Some of this multi-model mean difference is likely due to lower emissions of aerosol precursors in 2014 relative to 2000 along with updated historical estimates for CMIP6 (Hoesly et al., 2018; Lamarque et al., 2010), although it is not clear that this explains the reduction in model spread in CMIP6. It should also be borne in mind that our range does not include the E3SM model, which diagnosed aerosol forcing to be $-1.65$ W m$^{-2}$ for 2005–2014 from a pair of parallel all-forcing and pre-industrial aerosol

forcing atmosphere-only runs (fig. 25 in Golaz et al., 2019). This highlights the likelihood that the inclusion of more models
submitting results to RFMIP would extend the CMIP6 range of aerosol forcing, but the same may also have been true in CMIP5
where only a subset of models performed the sstClimAerosol experiment.

Atmospheric adjustments are small in magnitude in the aerosol forcing experiment, but large enough such that there is a
noticeable difference between ERF and RF (fig. 1; Table S3). The small non-cloud adjustments in most models shows that
the aerosol forcing is dominated by scattering aerosols (sulfate, organics, and for a limited number of models, nitrates) rather
than black carbon (Smith et al., 2018b). Additionally, in two of the four models that provide the single-forcing BC experiment
in AerChemMIP (CNRM-ESM2-1 and UKESM1-0-LL) the overall adjustment is small (Thornhill et al., 2020), in contrast to
findings in PDRMIP models (Smith et al., 2018b). In MRI-ESM2-0 (model 15) there are strong tropospheric temperature and
cloud changes to black carbon forcing resulting in a negative adjustment overall (Thornhill et al., 2020).

For aerosol forcing, the aerosol-cloud interactions dominate, with an increase in cloud optical depth at all cloud heights.
As cloud droplet effective radius decreases, cloud albedo, and hence optical depth, increases. This also implies that absorbing
aerosols play only a minor role in most models, as BC induces strong adjustments that cause a general increase in cloud height
in PDRMIP models from an increasing tropospheric stability (Smith et al. (2018b); Stjern et al. (2017); fig. S2). There is no
evidence of this in the RFMIP aerosol forcing experiment, although some models do also include aerosol-cloud interactions
from BC, and the effect may be due to the BC forcing being a smaller fraction of the total aerosol forcing than sulfate (Thornhill
et al., 2020). Figure S2 shows ISCCP simulator results for the five PDRMIP experiments from the CMIP5-era HadGEM2-ES
model, where it can be seen that the aerosol forcing experiment is qualitatively more similar to the $5{\times}SO_4$ forcing experiment
than the $10{\times}BC$ experiment in PDRMIP. The increase in cloud albedo leads to a strong negative SW radiative effect that is
partially compensated by LW effects (note that the ISCCP simulator kernel does not distinguish RFaci from adjustments).

Unlike for WMGHGs, aerosol forcing adjustments are dominated by cloud effects with only small non-cloud components
(table 5). For aerosol forcing, all model years are used, as the stratospheric temperature adjustment is negligible. The spread
in values of cloud adjustments is large, and spans positive and negative values. This reconfirms that atmospheric processes in
response to aerosol forcing remains one of the largest uncertainties in climate models. There is also a spread in tropospheric
temperature and water vapour adjustments with multi-model means near zero, suggesting that some models respond to aerosols
with substantial atmospheric warming or cooling.

For many regions, particularly Southern Asia and the Eastern Pacific, the aerosol ERF is driven by large and negative cloud
changes (fig. 8). The small adjustment overall and increase in cloud optical depth for all ISCCP cloud categories suggests this
is driven by an increase in cloud condensation nuclei leading to a more negative RFaci. There are some regions such as the
Sahara in which a positive ERF arises and not easily explained by any adjustment component. This may be a reduction in
mineral dust loading and increase in BC loading, leading to a positive forcing (e.g. as seen in NorESM2-LM, fig. S3).

The total derived cloud adjustment for aerosols is $-0.20$ W m$^{-2}$, derived of $-0.04$ W m$^{-2}$ from SW cloud liquid water path
adjustment, $-0.13$ W m$^{-2}$ from SW cloud fraction change, and $-0.03$ W m$^{-2}$ from cloud changes in the LW (table S7).

**Table 5.** As for table 3 but for 1850–2014 aerosol forcing.

| # | Model | ERF | IRF | Adj. | ts | ta_tr | ta_st | hus | albedo | cl |
|---|---|---|---|---|---|---|---|---|---|---|
| 1 | ACCESS-CM2 | −1.09 | | | 0.07 | 0.11 | 0.01 | −0.00 | −0.03 | |
| 2 | CanESM5 | −0.85 | −0.51 | −0.34 | 0.02 | −0.16 | −0.10 | 0.18 | 0.01 | −0.28 |
| 3 | CESM2 | −1.37 | −1.43 | 0.06 | −0.02 | −0.00 | −0.07 | 0.10 | −0.01 | 0.05 |
| 4 | CNRM-CM6-1 | −1.15 | −1.19 | 0.04 | 0.07 | 0.15 | −0.00 | −0.08 | −0.06 | −0.03 |
| 5 | CNRM-ESM2-1 | −0.74 | −0.75 | 0.01 | 0.06 | 0.10 | −0.01 | −0.06 | −0.02 | −0.06 |
| 6 | EC-Earth3 | −0.80 | −0.66 | −0.14 | 0.06 | −0.02 | 0.01 | −0.02 | −0.04 | −0.14 |
| 7 | GFDL-CM4 | −0.73 | −0.56 | −0.17 | 0.05 | −0.03 | −0.02 | 0.07 | −0.07 | −0.17 |
| 8 | GFDL-ESM4 | −0.70 | −0.37 | −0.33 | 0.04 | −0.15 | −0.01 | 0.10 | −0.05 | −0.26 |
| 9 | GISS-E2-1-G p1 | −1.32 | −0.46 | −0.86 | 0.06 | 0.22 | −0.03 | −0.09 | −0.09 | −0.93 |
| 10 | GISS-E2-1-G p3 | −0.93 | −1.00 | 0.07 | 0.05 | 0.13 | −0.03 | −0.06 | −0.00 | −0.01 |
| 11 | HadGEM3-GC31-LL | −1.10 | −1.04 | −0.06 | 0.05 | 0.10 | 0.01 | −0.05 | −0.04 | −0.14 |
| 12 | IPSL-CM6A-LR | −0.63 | −0.60 | −0.03 | 0.05 | 0.06 | −0.11 | 0.01 | −0.00 | −0.03 |
| 13 | MIROC6 | −1.04 | −1.13 | 0.10 | 0.06 | 0.17 | 0.01 | −0.10 | −0.03 | −0.02 |
| 15 | MRI-ESM2-0 | −1.21 | −0.46 | −0.74 | 0.04 | −0.24 | −0.00 | 0.17 | −0.02 | −0.68 |
| 16 | NorESM2-LM | −1.21 | −1.09 | −0.11 | 0.00 | 0.03 | −0.05 | 0.02 | −0.04 | −0.08 |
| 17 | NorESM2-MM | −1.26 | −1.10 | −0.16 | 0.03 | 0.01 | −0.02 | 0.05 | −0.04 | −0.19 |
| 18 | UKESM1-0-LL | −1.11 | −0.97 | −0.14 | 0.06 | 0.01 | 0.01 | 0.01 | −0.05 | −0.18 |
| | Mean | −1.01 | −0.83 | −0.18 | 0.05 | 0.03 | −0.02 | 0.01 | −0.03 | −0.20 |
| | St. dev. | 0.23 | 0.31 | 0.27 | 0.02 | 0.12 | 0.04 | 0.08 | 0.03 | 0.25 |

### 5.3.2 Relationship to climate sensitivity

The increase in the upper bound, and in the overall spread, of ECS in the CMIP6 model population compared to CMIP5 is well-documented (Forster et al., 2020; Zelinka et al., 2020). Figure 9 shows the relationships between ECS and transient climate response (TCR) and aerosol ERF in CMIP6, taking ECS and TCR from each model's abrupt4xCO2 and 1pctCO2 CMIP runs respectively. There are weak and non-significant positive correlations between ECS and aerosol forcing ($r = 0.12$) and between TCR and aerosol forcing ($r = 0.26$). This suggests that, as a population, models with high sensitivity are not tuning present-day aerosol forcing to be strong in order to reproduce observed warming[1]: it would be expected that these correlations would be negative if this was the case (Smith et al., 2018a). In CMIP5 models, aerosol forcing was stronger in models with higher ECS and TCR, but not significantly so (Forster et al., 2013), although significance emerges if one considers only models which include an aerosol indirect effect (Chylek et al., 2016). In CMIP3 there was a strong and significant negative

---

[1]MPI-ESM1-2 (Mauritsen et al., 2019) is the only documented exception. MIROC6 (Tatebe et al., 2019) did tune the aerosol forcing to better correspond to the AR5 best estimate but explicitly did not tune for surface temperature.

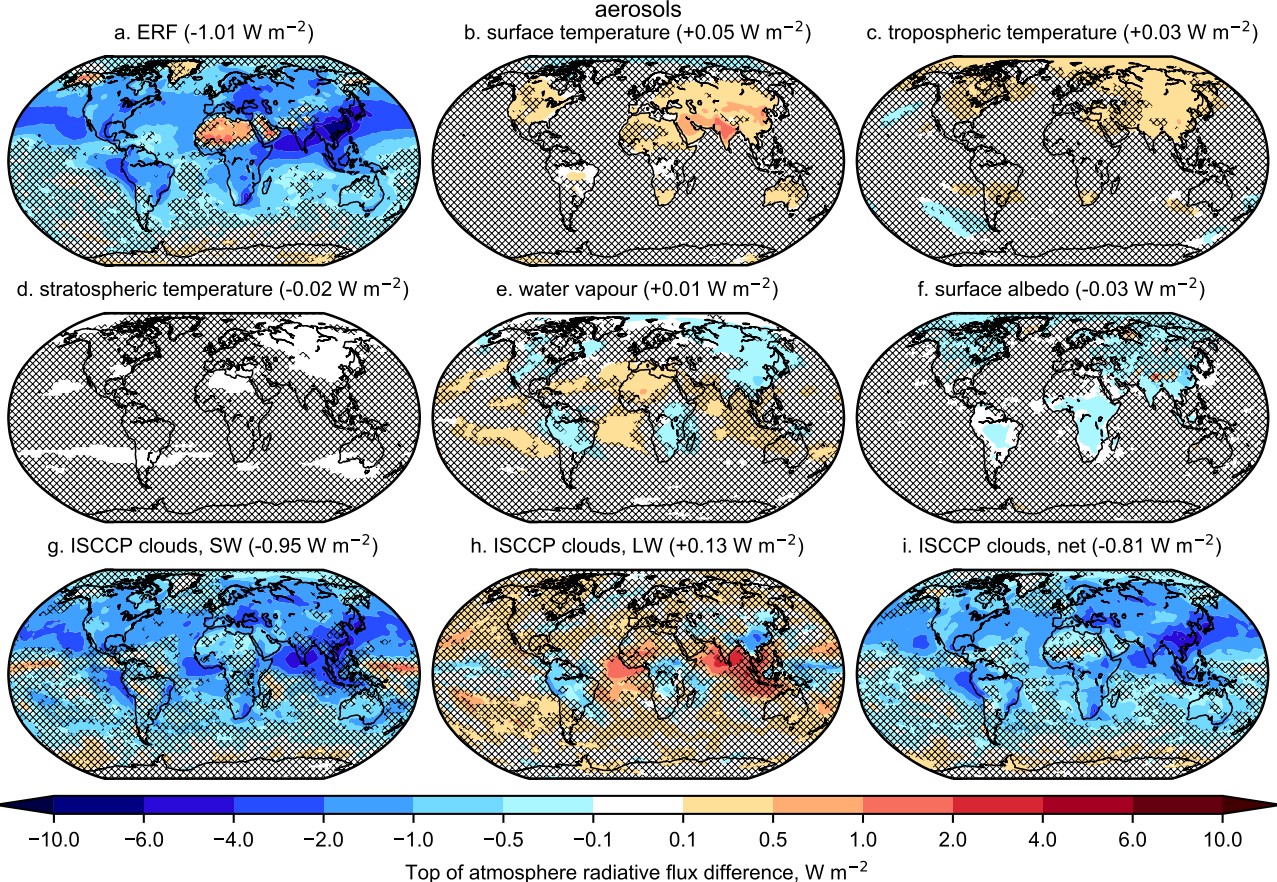

**Figure 8.** As fig. 6 but for present-day aerosol forcing.

correlation between climate sensitivity and aerosol forcing (Kiehl, 2007). It may be the case that aerosol forcing over some historical periods is stronger in CMIP6 than in CMIP5, as despite higher climate sensitivity, CMIP6 models warm less than CMIP5 models and observations up until 2000 (Flynn and Mauritsen, 2020).

### 5.3.3 Decomposition of aerosol forcing into aerosol-radiation and aerosol-cloud effects

The approximate partial radiative perturbation (APRP) method (Taylor et al., 2007) can be used to decompose shortwave (SW) aerosol forcing into aerosol-radiation interactions (ERFari), aerosol-cloud interactions (ERFaci), and the surface albedo adjustment (Zelinka et al., 2014). In section 5.3.4 we compare other methods to estimate ERFari and ERFaci. ERFari is the component of aerosol forcing that arises from the direct radiative effect of aerosol absorption and scattering (RFari) plus any adjustments (formerly known as the semi-direct effect) arising from perturbations in tropospheric heating rates, humidity, and their consequential effects on where clouds form (Boucher et al., 2013). These adjustments tend to be strong for black carbon but weak for scattering aerosol (Smith et al., 2018b; Stjern et al., 2017). ERFaci is composed of any changes in cloud albedo

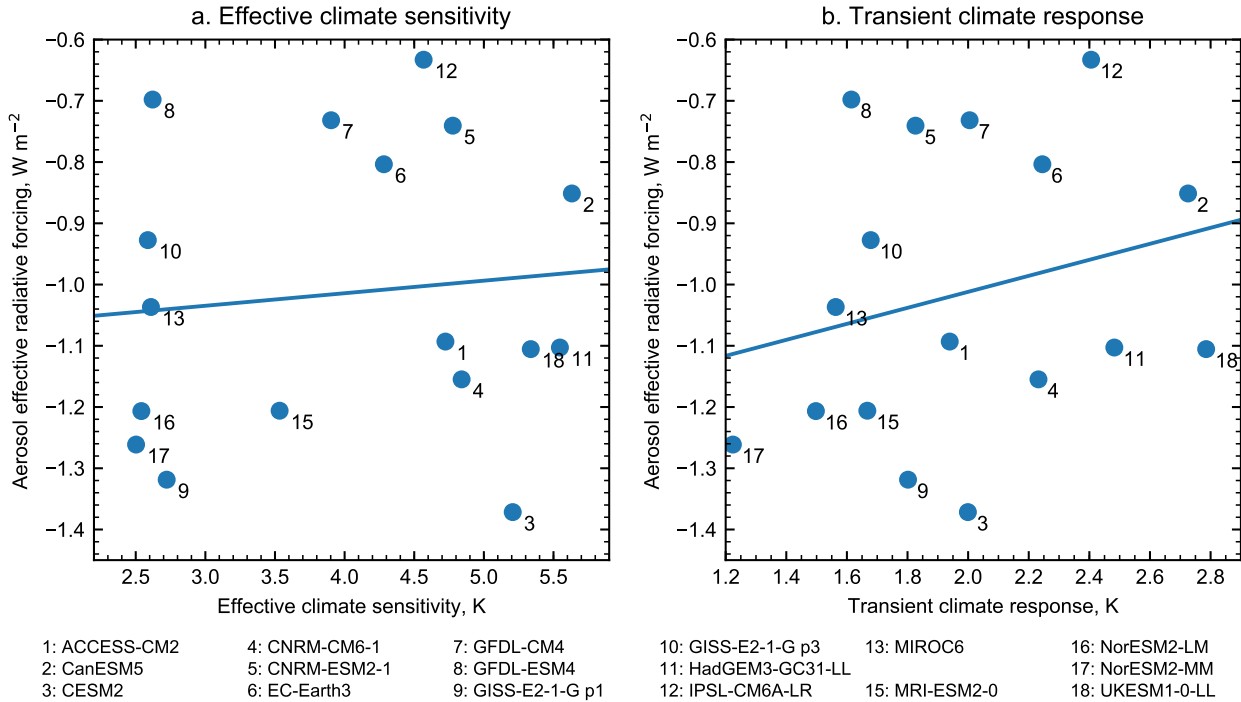

**Figure 9.** Relationship between (a) ECS and (b) TCR and aerosol ERF in the CMIP6 model ensemble. MPI-ESM1.2-LR (model 14) did not produce the piClim-aer experiment.

resulting from aerosols acting as cloud condensation nuclei and changing cloud droplet effective radius (RFaci, formerly the first indirect or Twomey effect, Twomey (1977)), plus adjustments relating to cloud lifetime and precipitation efficiency that changes liquid water path and cloud fraction (formerly second indirect or Albrecht effect, Albrecht (1989)). RFaci tends to be strong for sulfate aerosol, but several models also include cloud interactions to other aerosol species, and four models (CESM2, MIROC6, MRI-ESM2-0 and NorESM2-LM) include aerosol interaction on ice clouds. The direct plus Twomey effects (RFari+aci) are treated as the IRF component of aerosol forcing, with the remaining components of ERFari+aci as adjustments (Boucher et al., 2013).

There is no equivalent longwave (LW) method to APRP, so we take the approach of Zelinka et al. (2014) and use the cloud radiative effect to decompose LW ERF into ERFari and ERFaci. The advantages of these techniques are that they only require standard CMIP output, and all participating models can provide estimates. Results are displayed in table 6 and shown in fig. 10. In table S8 the equivalent SW ERFari for clear sky conditions are shown.

The total ERFari+aci from the APRP method is $-1.04$ ($\pm$ 0.20) W m$^{-2}$, agreeing very well with the ERF estimate of $-1.01$ ($\pm$0.23) W m$^{-2}$. ERFari+aci is approximately 22% from ERFari and 78% from ERFaci, and is comprised of a SW contribution of $-1.26$ W m$^{-2}$ offset by a LW contribution of $+0.23$ W m$^{-2}$. The model spread in both the SW and LW individual components is larger than for the net forcing. This is driven by the four models that include ice cloud interactions

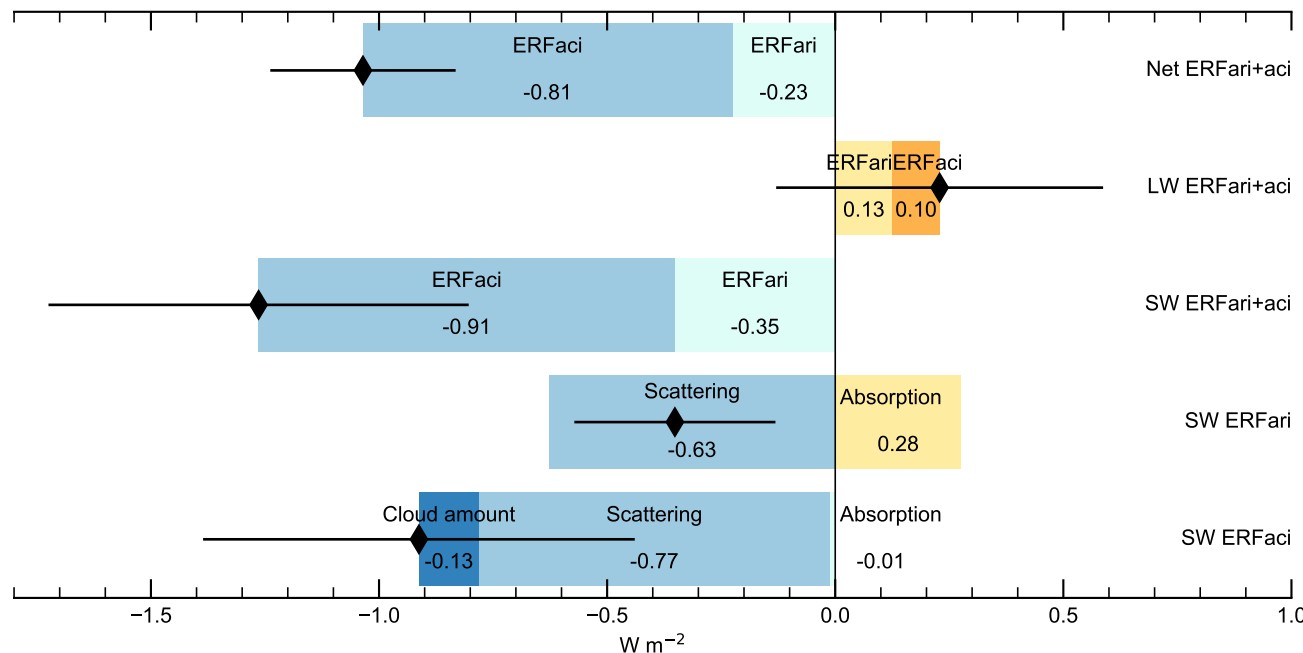

**Figure 10.** Components of the aerosol forcing diagnosed from the Approximate Partial Radiative Perturbation (for SW aerosol components) and from the cloud radiative effect (for LW components). Black diamonds represent multi-model means, black bars show one standard deviation.

that show positive LW ERFaci offset by strong negative SW ERFaci. MRI-ESM2.0 in particular has a very large positive LW ERFaci of $+1.47$ W m$^{-2}$, which comes from ice cloud nucleation by black carbon aerosols with temperature below $-38°$C in high-level clouds in the tropics (Oshima et al., in prep.). For the SW component the ERFari/ERFaci split is approximately 28% to 72%.

Multi-model mean SW ERFari is $-0.35$ W m$^{-2}$, comprised of an absorption of $+0.28$ W m$^{-2}$ offset by scattering of $-0.63$ W m$^{-2}$. The SW ERFaci is $-0.91$ W m$^{-2}$, made up of scattering ($-0.77$ W m$^{-2}$), absorption ($-0.01$ W m$^{-2}$) and cloud fraction change ($-0.13$ W m$^{-2}$).

### 5.3.4 Comparison of ERFari and ERFaci methods

Eight models also archived radiation diagnostics from aerosol-free radiation calls (the double call method) as recommended by Ghan (2013) which allows separation into ERFari and ERFaci. This can be compared with the APRP estimates in the SW and cloud radiative effect for the LW. Figure 11 shows different methods of estimating ERFaci and ERFari from the aerosol forcing experiment. For ERFaci in both the SW and LW, different methods provide similar estimates. For ERFari, the APRP and double-call methods sometimes disagree on sign for SW forcing, but this component is relatively small compared to the

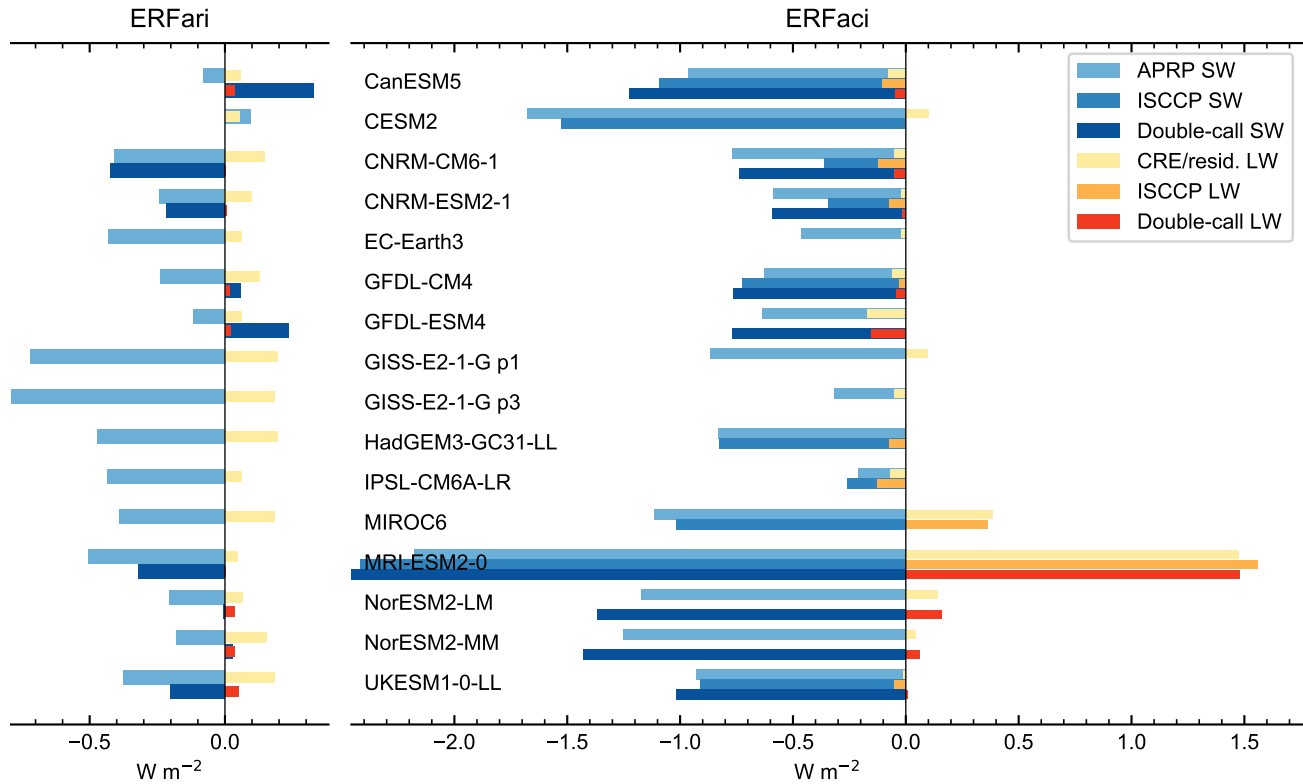

**Figure 11.** Comparison of methods to estimate ERFaci and ERFari from the aerosol experiment. CRE/resid. is the LW cloud radiative effect for ERFaci and the difference of LW ERF and CRE for ERFari. Not all methods are available in all models.

SW ERFaci where estimates are generally more consistent between APRP and the double call. Similarly in the LW, the CRE and double call methods produce similar results for ERFaci with larger relative differences for the smaller ERFari component. The double-call method is considered to be quite reliable, but the good agreement between the APRP or CRE and double call methods suggest that these simpler tools are useful tool to diagnose ERFari and ERFaci from climate models, which is advantageous due to there being no requirement for specialised model diagnostics.

## 5.4 Land-use change

Land use ERF is small and not significant at $-0.09$ ($\pm 0.13$) W m$^{-2}$. Forcing and adjustments are difficult to distinguish from zero and it is unlikely that this forcing played a large role historically for global mean impacts. In 13 of the 14 models that ran this experiment, land-use ERF is negative, and the multi-model mean and standard deviation is affected by a relatively large positive forcing in the NorESM2-LM model. In fig. S3g we show that this is due to cloud adjustments in this model. This is a consequence of interactive isoprene and monoterpene specified from the land surface changes, causing a reduction in organic

matter, reducing cloud condensation nuclei and increasing SW cloud adjustment (unlike for the aerosol forcing experiment, the Twomey effect in response to a land use forcing is treated as an adjustment and not a forcing, because anthropogenic aerosol emissions are not perturbed). In other models, where the ERF is small and negative, it should also be borne in mind that internal variability may make it more difficult to isolate the forcing signal from the noise in free-running simulations (Forster et al., 2016), although the multi-model mean is likely to be more robust than individual model results. This experiment was partly motivated by a large land use forcing of $-0.4$ W m$^{-2}$ in the CMIP5 HadGEM2-ES model (Andrews et al., 2017), which showed a large change in regional dust loading that contributed to this forcing. Our multi-model mean ERF of $-0.09$ W m$^{-2}$ ($-0.12$ W m$^{-2}$ if NorESM2-LM is excluded) agrees well with an observational-constrained analysis from CMIP5 models of $-0.11$ W m$^{-2}$ (Lejeune et al., 2020), and is within the likely range of the AR5 assessment of $-0.15$ ($-0.05$ to $-0.25$) W m$^{-2}$.

The radiative forcing from land use change is driven by the resulting change in surface albedo. For example, deforestation for agricultural use converts relatively dark forest cover to brighter cropland, exerting a negative forcing (Betts, 2000, 2001). The surface albedo kernel-derived flux change is taken to be the IRF. It is not a perfect measure as it includes changes in snow and ice cover over land, and any biophysical response, as both changes in land surface temperatures and surface properties can affect snow cover. However, the land surface temperature change is very small in the land-use experiment, evidenced by the small land surface temperature adjustment in fig. 3. In Fig. S4 we show changes in aerosol optical depth at 550 nm for models that provided this diagnostic. There is diversity in the model aerosol loadings to land-use forcing that does not appear to explain the diversity in land-use ERF between models. In particular, CanESM5 has a strong aerosol optical depth increase to land-use change, but a relatively weak ERF of $-0.08$ W m$^{-2}$. Changes in surface properties such as how snow cover settles over different land types and the biophysical response is not easy to discern from model output. Again, all available model years are used because stratospheric temperature adjustment does not play a large role.

The spatial pattern of land-use forcing and adjustments (fig. 12) is generally not significant in many parts of the world due to the small size of the forcing. The exception to this is water vapour and SW cloud adjustments over the Amazon; deforestation from pre-industrial to present-day is likely to have reduced evapotranspiration from vegetation, reducing tropospheric humidity and low level cloud cover. These spatial patterns are also coincident with a decrease in organic carbon loading in NorESM2-LM (fig. S3).

## 5.5 Anthropogenic total

The total anthropogenic ERF for 1850–2014 stands at 2.00 ($\pm$ 0.23) W m$^{-2}$. Inter-model spread is larger, both in relative and absolute terms, in the total anthropogenic forcing than it is for any of its individual components, suggesting that individual models respond very differently to the same combinations of forcing. In the absence of non-linearities between forcing components, the residual ERF of $+0.21$ W m$^{-2}$ from the land-use, aerosol and WMGHG components compared to the total anthropogenic would mostly be comprised of ozone forcing, although the sum of individual forcings does not necessarily equal the total forcing in some models (Thornhill et al., 2020; O'Connor et al., 2020). As for the aerosol forcing experiment, there is no significant correlation between total anthropogenic forcing and ECS or TCR.

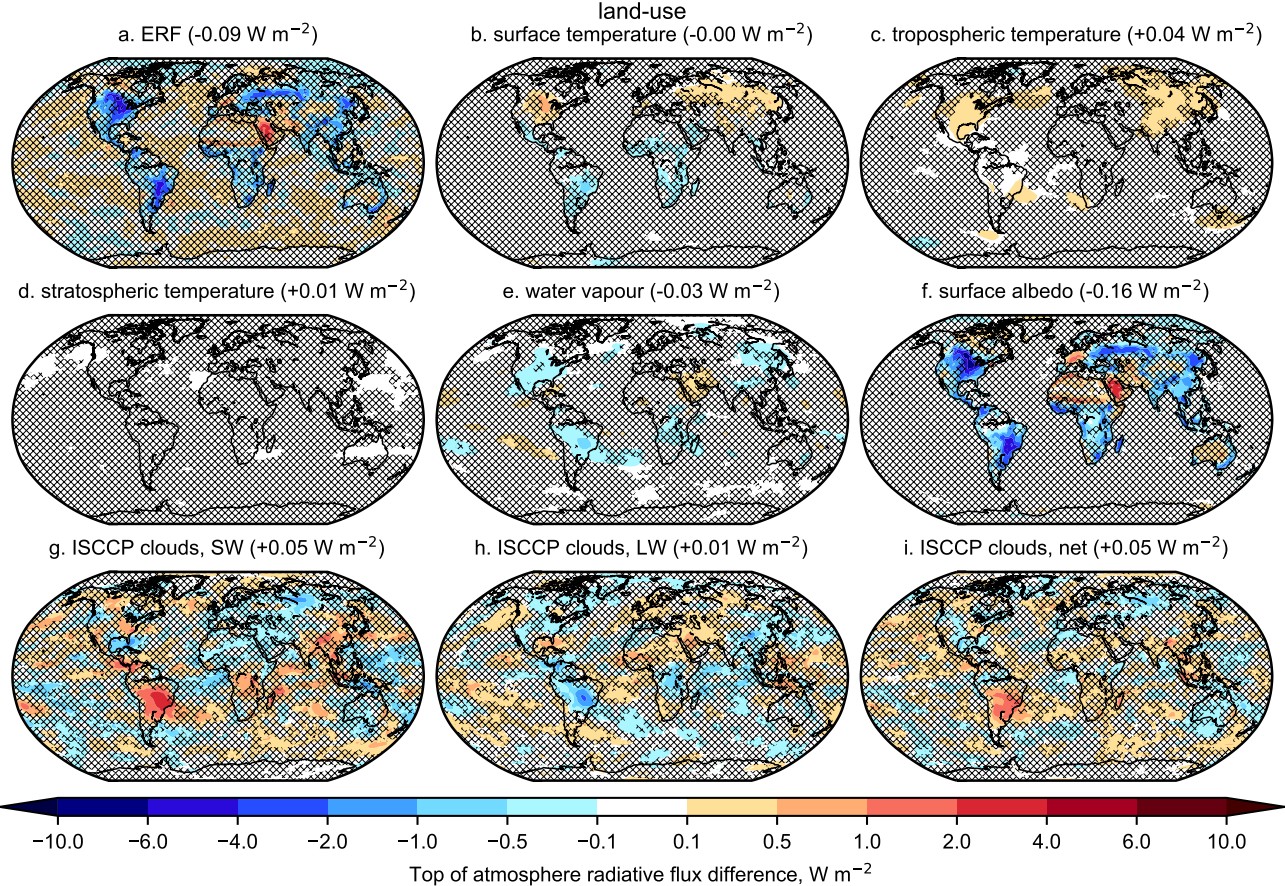

**Figure 12.** As fig. 6 but for present-day land use forcing. Subplot (f) represents the IRF in this experiment.

The total anthropogenic forcing shows the offsetting influences of the greenhouse gas and aerosol forcing components on the ERF, IRF and adjustments. The total anthropogenic ISCCP-simulator cloud changes are also a combination of the WMGHG and aerosol contributions, with the net effect being dominated by aerosol. For non-cloud adjustments the combination of strong positive adjustments from the greenhouse gas forcing with a small negative adjustment from the aerosol forcing results in an adjustment that is of comparable magnitude to the IRF. The exception is the GISS-E2-1-G model r1i1p1f1 variant that has a very strong negative cloud adjustment, driven by a large increase in cloud fraction in the aerosol experiment (table 6). Additionally, stratospheric adjustment is stronger for total anthropogenic forcing than for WMGHGs alone, suggesting a role for ozone forcing in contributing to this adjustment.

The pattern of anthropogenic forcing is spatially inhomogeneous; positive where aerosol forcing is weak, and negative where localised aerosol-cloud effects dominate (fig. 13). The influence of WMGHG forcing on temperature and water vapour adjustments, and of aerosol forcing on the cloud response, is evident. For all forcings, but particularly for land-use, aerosol and total anthropogenic, many of the forcing and adjustment terms do not show robust signals regionally. This indicates

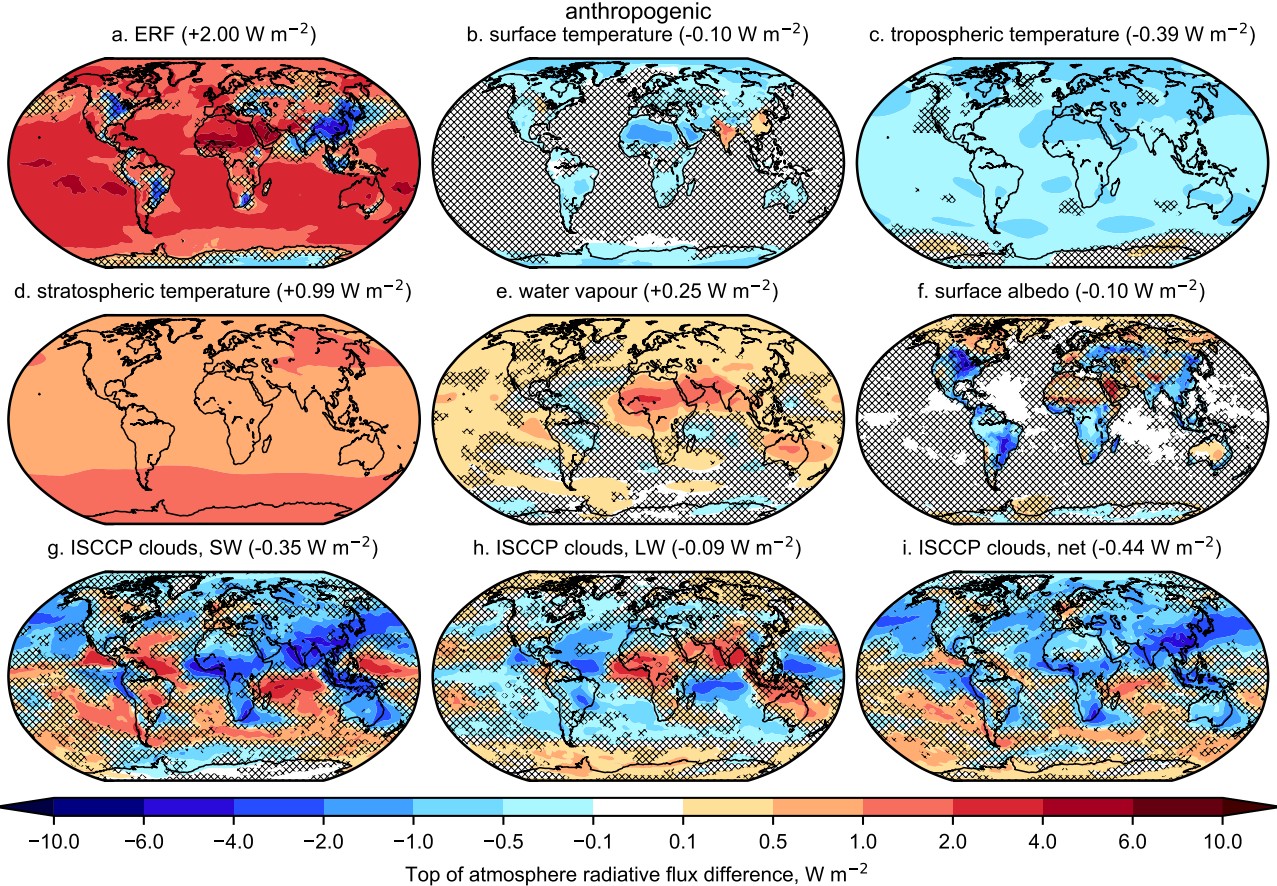

anthropogenic

a. ERF (+2.00 W m$^{-2}$)  b. surface temperature (-0.10 W m$^{-2}$)  c. tropospheric temperature (-0.39 W m$^{-2}$)

d. stratospheric temperature (+0.99 W m$^{-2}$)  e. water vapour (+0.25 W m$^{-2}$)  f. surface albedo (-0.10 W m$^{-2}$)

g. ISCCP clouds, SW (-0.35 W m$^{-2}$)  h. ISCCP clouds, LW (-0.09 W m$^{-2}$)  i. ISCCP clouds, net (-0.44 W m$^{-2}$)

−10.0  −6.0  −4.0  −2.0  −1.0  −0.5  −0.1  0.1  0.5  1.0  2.0  4.0  6.0  10.0

Top of atmosphere radiative flux difference, W m$^{-2}$

**Figure 13.** As fig. 6 but for present-day anthropogenic forcing.

that adjustments are best considered as global-mean quantities that affect the globally-resolved forcing-feedback framework (eq. (1)).

## 6  Conclusions

Effective radiative forcing is the driving process behind long-term changes in global mean surface temperature. As ERF is now preferred to RF, climate models are the best tools we have to determine the heating impacts of various species on the Earth atmosphere system.

From CMIP5 to CMIP6, both $CO_2$ and aerosol forcing has become more consistent across the population of participating models. This has helped to address a concern from CMIP5: that forcing was poorly characterised in CMIP5 models and

inconsistently determined (Stouffer et al., 2017). Multi-model mean $CO_2$ and all-WMGHG ERF estimates agree very well with RF estimates from Etminan et al. (2016) using a line-by-line radiative transfer model (sections 5.1 and 5.2). A comprehensive

review of aerosol forcing placed the 16–84% uncertainty range in present-day aerosol ERF at $-1.60$ to $-0.65$ W m$^{-2}$ (Bellouin et al., 2020b). Results from CMIP6 models show a relatively tight spread of $-1.37$ to $-0.63$ W m$^{-2}$ for the full range. Although 17 models is a reasonable sample size of the CMIP6 population, more models may submit forcing results to CMIP6 that would widen this range (and indeed, we would encourage modelling groups to do so). One example is E3SM which did not perform the RFMIP aerosol forcing experiment but where it would be likely that the 1850–2014 aerosol forcing would be more negative than $-1.37$ W m$^{-2}$ (fig. 25 in Golaz et al., 2019). While the increase in $4\times$CO$_2$ forcing compared to CMIP5 may explain some of the increase in climate sensitivity in CMIP6 models (Zelinka et al., 2020), the model range of present-day aerosol forcing does not, particularly for the upper bound.

We determine a multi-model mean anthropogenic ERF of 2.00 ($\pm$0.23) W m$^{-2}$ for 1850–2014. This is less than the anthropogenic ERF in AR5 for 1850–2011 of 2.24 W m$^{-2}$ (Myhre et al. (2013); although this figure has a wide uncertainty range), and extrapolating trends forward would suggest an anthropogenic ERF of around 2.4 W m$^{-2}$ from AR5 for 1850–2014. The two main reasons for this difference are a stronger negative aerosol forcing in CMIP6 compared to the AR5 assessment ($-1.01$ W m$^{-2}$ in CMIP6 for 1850–2014 versus $-0.72$ W m$^{-2}$ in AR5 for 1850–2011), and a weaker ozone forcing ($+0.21$ W m$^{-2}$ versus $+0.31$ W m$^{-2}$), if residual anthropogenic forcing is attributed to ozone.

Forcing adjustments produce insight into the atmospheric mechanisms that contribute to ERF. Warming of the troposphere results in a negative adjustment due to the increase in outgoing LW radiation, and increasing water vapour counteracts this effect partially by its role as a greenhouse gas. All models agree on tropospheric warming and moistening for WMGHG and all anthropogenic forcing. These tropospheric adjustments are small for aerosol forcing but models do not agree on the sign of the change. The instantaneous radiative forcing and cloud adjustments are generally the largest sources of inter-model spread in the forcing component in climate models. Since IRF is not directly calculated in this study, some of this spread may be from residuals in the kernel decomposition and the true spread in IRF may be smaller than reported here. One strand of RFMIP will include benchmarking of GCM radiative transfer against line-by-line codes. Radiative transfer is a well-grounded theoretical problem where the diversity in line-by-line codes is small (Pincus et al., 2015), so this component of inter-model diversity has a measurable yardstick for improvement. Cloud responses are more difficult to constrain and exhibit a wide range of behaviour to both greenhouse gas and aerosol forcing. However, progress is beginning to be made. For greenhouse gas forcing, techniques from the climate feedback literature that have observational parallels, such as analysing cloud-controlling factors (Klein et al., 2017), can be applied to adjustments. Use of the ISCCP simulator diagnostics with the ISCCP cloud kernel, another method conceptualised by climate feedback investigations (Zelinka et al., 2012), allows cloud adjustments to be calculated directly facilitating better inter-model comparison. For aerosol forcing, observational methods exist to determine RFari and RFaci using satellite and reanalysis data (Bellouin et al., 2013, 2020a). Ultimately, reducing uncertainty in effective radiative forcing will reduce uncertainty in climate projections due to the central role of forcing in driving the Earth's global mean temperature response.

*Data availability.*  RFMIP model data used in this study is freely available from the CMIP6 repository on the Earth System Grid Foundation
nodes (https://esgf-node.llnl.gov/search/cmip6/). The HadGEM3-GA7.1 kernels are available at https://doi.org/10.5281/zenodo.3594673.

*Author contributions.*  C.J.S. co-ordinated the project, analysed the data and led the writing of the manuscript. R.J.K. produced adjustment
calculations using radiative kernels. G.M. provided adjustment calculations using offline radiation simulations. K.A. provided model results
of effective radiative forcing and effective climate sensitivity. W.C. provided analysis of different definitions of effective radiative forcing.
R.P. and P.M.F. oversaw the RFMIP-ERF project in coordination with the World Climate Research Programme under the CMIP6 protocol.
All other authors ran models and published results on the Earth System Grid Foundation, without which this work would not have been
possible. All authors contributed to the writing and review.

*Competing interests.*  The authors declare no competing interests.

*Disclaimer.*  TEXT

*Acknowledgements.*  We thank Ed Gryspeerdt, Mark Zelinka, Roland Séférian and the participants and hosts of the Tri-MIP-athlon con-
sortium meetings in Reading 2018 and Princeton 2019 for fruitful discussions. We acknowledge the World Climate Research Programme,
which, through its Working Group on Coupled Modelling, coordinated and promoted CMIP6. We thank the climate modeling groups for
producing and making available their model output, the Earth System Grid Federation (ESGF) for archiving the data and providing access,
and the multiple funding agencies who support CMIP6 and ESGF.

   C.J.S. was supported by a NERC/IIASA Collaborative Research Fellowship (NE/T009381/1). C.J.S. and P.M.F. were supported by the
European Union's Horizon 2020 research and innovation programme under grant agreement No 820829 (CONSTRAIN project). R.J.K. is
supported by an appointment to the NASA Postdoctoral Program at NASA Goddard Space Flight Center. R.P. was supported by the US
Department of Energy's Office of Biological and Environmental Research under grant 7457436 from Lawrence Berkeley National Lab.
T.A., F.M.O'C., E.R., and A.W. were supported by the Met Office Hadley Centre Climate Programme funded by BEIS and Defra and the
Newton Fund through the Met Office Climate Science for Service Partnership Brazil (CSSP Brazil). F.M.O'C. also acknowledges support
from the EU Horizon 2020 Research Programme CRESCENDO project, grant agreement number 641816. A.K. and D.O. were supported
by the Research Council of Norway (grant nos. 229771, 285003, and 285013), by Notur/NorStore (NN2345K and NS2345K), and through
EU H2020 grant no. 280060. H.S. was supported by TOUGOU (MEXT, Japan). The MIROC6 simulations were performed using the Earth
Simulator at JAMSTEC and the NEC SX at NIES. The CMIP6 project at IPSL used the HPC resources of TGCC under the allocations
2017-R0040110492 and 2018-R0040110492 (project gencmip6) provided by GENCI (Grand Équipement National de Calcul Intensif). The
CESM project is supported primarily by the National Science Foundation. Part of this material is based upon work supported by the National
Center for Atmospheric Research, which is a major facility sponsored by the National Science Foundation under Cooperative Agreement
No. 1852977. The simulations with EC-Earth3 were performed on resources provided by the Swedish National Infrastructure for Computing

(SNIC) at the National Supercomputer Centre (NSC) partially funded by the Swedish Research Council through grant agreement no. 2016-07213. M.D. and C.M. were supported by funding from the Earth Systems and Climate Change Hub of the Australian Government's National Environmental Science Program. ACCESS modelling was undertaken with the assistance of resources from the National Computational Infrastructure (NCI Australia), an NCRIS enabled capability supported by the Australian Government.

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

**Table 6.** Contribution of the components of effective radiative forcing from the present-day aerosol time-slice RFMIP experiment.

| Model | SW ARI scat | SW ARI abs | SW ARI sum | SW ACI scat | SW ACI abs | SW ACI amt | SW ACI sum | SW ARI+ACI | LW ARI | LW ACI | LW ARI+ACI | Net ARI | Net ACI | Net ARI+ACI |
|---|---|---|---|---|---|---|---|---|---|---|---|---|---|---|
| ACCESS-CM2 | −0.79 | 0.30 | −0.48 | −0.80 | −0.01 | −0.12 | −0.93 | −1.42 | 0.26 | 0.04 | 0.30 | −0.22 | −0.89 | −1.11 |
| CanESM5 | −0.60 | 0.52 | −0.08 | −0.88 | 0.06 | −0.14 | −0.96 | −1.04 | 0.06 | −0.08 | −0.02 | −0.02 | −1.04 | −1.06 |
| CESM2 | −0.26 | 0.35 | 0.09 | −1.70 | 0.03 | −0.01 | −1.68 | −1.58 | 0.05 | 0.10 | 0.16 | 0.15 | −1.57 | −1.43 |
| CNRM-CM6-1 | −0.61 | 0.20 | −0.41 | −0.77 | −0.05 | 0.05 | −0.77 | −1.17 | 0.15 | −0.05 | 0.09 | −0.26 | −0.82 | −1.08 |
| CNRM-ESM2-1 | −0.42 | 0.18 | −0.24 | −0.52 | −0.04 | −0.03 | −0.59 | −0.83 | 0.10 | −0.02 | 0.08 | −0.14 | −0.61 | −0.75 |
| EC-Earth3 | −0.75 | 0.32 | −0.43 | −0.34 | −0.04 | −0.08 | −0.46 | −0.89 | 0.06 | −0.02 | 0.04 | −0.37 | −0.48 | −0.85 |
| GFDL-CM4 | −0.65 | 0.41 | −0.24 | −0.54 | 0.00 | −0.09 | −0.62 | −0.86 | 0.13 | −0.06 | 0.07 | −0.11 | −0.68 | −0.80 |
| GFDL-ESM4 | −0.65 | 0.53 | −0.12 | −0.59 | 0.01 | −0.06 | −0.63 | −0.75 | 0.06 | −0.17 | −0.11 | −0.06 | −0.80 | −0.86 |
| GISS-E2-1-G p1 | −0.91 | 0.19 | −0.72 | 0.06 | 0.01 | −0.94 | −0.87 | −1.58 | 0.19 | 0.10 | 0.29 | −0.52 | −0.77 | −1.29 |
| GISS-E2-1-G p3 | −0.97 | 0.18 | −0.79 | −0.25 | −0.01 | −0.06 | −0.32 | −1.11 | 0.18 | −0.05 | 0.13 | −0.61 | −0.37 | −0.97 |
| HadGEM3-GC31-LL | −0.77 | 0.30 | −0.47 | −0.75 | −0.01 | −0.07 | −0.83 | −1.30 | 0.20 | −0.00 | 0.19 | −0.28 | −0.83 | −1.11 |
| IPSL-CM6A-LR | −0.60 | 0.17 | −0.43 | −0.26 | −0.01 | 0.06 | −0.21 | −0.65 | 0.06 | −0.07 | −0.01 | −0.37 | −0.28 | −0.65 |
| MIROC6 | −0.48 | 0.09 | −0.39 | −1.04 | −0.06 | −0.01 | −1.12 | −1.51 | 0.18 | 0.38 | 0.57 | −0.21 | −0.73 | −0.94 |
| MRI-ESM2-0 | −0.70 | 0.19 | −0.51 | −1.71 | −0.09 | −0.38 | −2.18 | −2.68 | 0.05 | 1.47 | 1.52 | −0.46 | −0.70 | −1.16 |
| NorESM2-LM | −0.42 | 0.21 | −0.21 | −1.07 | −0.00 | −0.10 | −1.17 | −1.38 | 0.07 | 0.14 | 0.21 | −0.14 | −1.03 | −1.17 |
| NorESM2-MM | −0.40 | 0.22 | −0.18 | −1.10 | 0.02 | −0.17 | −1.25 | −1.43 | 0.15 | 0.04 | 0.20 | −0.03 | −1.20 | −1.23 |
| UKESM1-0-LL | −0.71 | 0.34 | −0.37 | −0.82 | −0.01 | −0.10 | −0.93 | −1.30 | 0.18 | −0.01 | 0.17 | −0.19 | −0.94 | −1.13 |
| Mean | −0.63 | 0.28 | −0.35 | −0.77 | −0.01 | −0.13 | −0.91 | −1.26 | 0.13 | 0.10 | 0.23 | −0.23 | −0.81 | −1.04 |
| St. dev. | 0.18 | 0.12 | 0.22 | 0.46 | 0.04 | 0.22 | 0.47 | 0.46 | 0.06 | 0.36 | 0.36 | 0.19 | 0.30 | 0.20 |

**Table 7.** As for table 3 but for 1850–2014 land-use forcing.

| # | Model | ERF | IRF | Adj. | ts | ta_tr | ta_st | hus | cl |
|---|---|---|---|---|---|---|---|---|---|
| 2 | CanESM5 | −0.08 | −0.10 | 0.03 | 0.02 | −0.02 | 0.01 | −0.00 | 0.02 |
| 3 | CESM2 | −0.04 | −0.09 | 0.05 | −0.01 | 0.03 | 0.00 | 0.01 | 0.04 |
| 5 | CNRM-ESM2-1 | −0.07 | −0.09 | 0.03 | 0.01 | −0.02 | −0.01 | 0.04 | 0.00 |
| 6 | EC-Earth3 | −0.13 | −0.13 | −0.00 | 0.02 | 0.07 | 0.02 | −0.03 | −0.09 |
| 7 | GFDL-CM4 | −0.33 | −0.41 | 0.08 | −0.04 | 0.09 | 0.00 | −0.06 | 0.08 |
| 8 | GFDL-ESM4 | −0.28 | −0.27 | −0.01 | −0.03 | 0.08 | 0.01 | −0.11 | 0.04 |
| 9 | GISS-E2-1-G p1 | −0.00 | 0.02 | −0.02 | −0.02 | −0.02 | 0.01 | 0.02 | −0.01 |
| 11 | HadGEM3-GC31-LL | −0.11 | −0.18 | 0.07 | 0.01 | 0.10 | 0.01 | −0.05 | −0.00 |
| 12 | IPSL-CM6A-LR | −0.05 | −0.09 | 0.05 | −0.01 | 0.02 | 0.00 | −0.01 | 0.05 |
| 13 | MIROC6 | −0.03 | −0.07 | 0.04 | −0.01 | 0.04 | 0.00 | −0.04 | 0.04 |
| 14 | MPI-ESM1-2-LR | −0.10 | −0.06 | −0.04 | −0.01 | 0.01 | 0.01 | −0.01 | −0.05 |
| 15 | MRI-ESM2-0 | −0.17 | −0.32 | 0.15 | 0.00 | 0.08 | −0.00 | −0.08 | 0.15 |
| 16 | NorESM2-LM | 0.26 | −0.00 | 0.27 | 0.01 | 0.01 | 0.00 | 0.00 | 0.24 |
| 18 | UKESM1-0-LL | −0.18 | −0.18 | 0.00 | 0.00 | 0.04 | 0.01 | −0.05 | −0.00 |
| | Mean | −0.09 | −0.14 | 0.05 | −0.00 | 0.04 | 0.01 | −0.03 | 0.04 |
| | St. dev. | 0.13 | 0.12 | 0.08 | 0.02 | 0.04 | 0.01 | 0.04 | 0.08 |

**Table 8.** As for table 3 but for 1850–2014 anthropogenic forcing.

| # | Model | ERF | IRF | Adj. | ts | ta_tr | ta_st | hus | albedo | cl |
|---|---|---|---|---|---|---|---|---|---|---|
| 1 | ACCESS-CM2 | 1.90 | | | −0.08 | −0.37 | 0.97 | 0.23 | −0.00 | |
| 2 | CanESM5 | 2.37 | 1.85 | 0.52 | −0.14 | −0.70 | 0.87 | 0.45 | 0.03 | 0.01 |
| 3 | CESM2 | 2.05 | 0.74 | 1.31 | −0.17 | −0.48 | 0.95 | 0.44 | 0.01 | 0.57 |
| 4 | CNRM-CM6-1 | 1.61 | 0.55 | 1.06 | −0.07 | −0.22 | 1.01 | 0.07 | 0.01 | 0.26 |
| 5 | CNRM-ESM2-1 | 1.66 | 0.67 | 1.00 | −0.04 | −0.32 | 1.11 | 0.16 | −0.03 | 0.12 |
| 6 | EC-Earth3 | 2.09 | | | −0.09 | −0.36 | 1.18 | 0.18 | −0.13 | |
| 7 | GFDL-CM4 | 2.34 | 2.16 | 0.18 | −0.11 | −0.44 | 0.69 | 0.31 | −0.36 | 0.10 |
| 8 | GFDL-ESM4 | 2.17 | 1.49 | 0.68 | −0.12 | −0.45 | 1.14 | 0.32 | −0.27 | 0.06 |
| 9 | GISS-E2-1-G p1 | 1.93 | 1.82 | 0.11 | −0.08 | −0.14 | 0.99 | 0.13 | 0.13 | −0.92 |
| 11 | HadGEM3-GC31-LL | 1.81 | 1.00 | 0.81 | −0.09 | −0.26 | 0.97 | 0.13 | −0.20 | 0.27 |
| 12 | IPSL-CM6A-LR | 2.32 | 1.41 | 0.91 | −0.11 | −0.40 | 0.94 | 0.28 | −0.08 | 0.28 |
| 13 | MIROC6 | 1.80 | 1.11 | 0.69 | −0.10 | −0.29 | 0.87 | 0.14 | −0.02 | 0.08 |
| 14 | MPI-ESM1-2-LR | 2.13 | | | −0.10 | −0.48 | 0.79 | 0.25 | 0.02 | |
| 15 | MRI-ESM2-0 | 1.95 | 1.77 | 0.18 | −0.10 | −0.59 | 0.93 | 0.44 | −0.28 | −0.22 |
| 16 | NorESM2-LM | 2.06 | 1.18 | 0.88 | −0.14 | −0.49 | 0.99 | 0.31 | −0.05 | 0.26 |
| 18 | UKESM1-0-LL | 1.79 | 0.63 | 1.16 | −0.10 | −0.28 | 1.24 | 0.11 | −0.20 | 0.39 |
| | Mean | 2.00 | 1.26 | 0.73 | −0.10 | −0.39 | 0.98 | 0.25 | −0.09 | 0.10 |
| | St. dev. | 0.23 | 0.51 | 0.37 | 0.03 | 0.14 | 0.14 | 0.12 | 0.13 | 0.35 |