# Peer review of "Effective radiative forcing and adjustments in CMIP6 models"

_Atmospheric Chemistry and Physics, 2019_

## Referee Comment (RC1) · Anonymous Referee #1 · 21 Feb 2020

General Comments:

The paper presents a comprehensive analysis of the diagnosed values of effective radiative forcing (ERF) for the CMIP6 models, and breaks down the contributions of this forcing from greenhouse gases, aerosols, and land-use. The use of ERF has continued to grow and it is now at least as widely-used, if not more so, than traditional metrics of forcing such as instantaneous forcing or stratospheric-adjusted forcing. It is an important paper for benchmarking the performance of CMIP6 models, and its findings will hopefully be used in upcoming assessment reports. That being said, there are many different findings in this paper and it would likely be more digestible if the findings were explored in more detail in separate papers. This is something of an omnibus paper. However, despite its size, the paper has important findings and only

requires minor revisions before being acceptable for publication.

Specific Comments:

The last point made by the authors in the abstract, which appears to be supported in Figure 8, namely that they see no evidence that aerosols are contributing to the spread in ECS, is striking and bears more discussing. The range and change in ECS for CMIP6 is and should be of great concern to the large numbers of individuals involved in CMIP6 and to the scientific community as a whole, since an explanation is required. I hope that a body of literature will emerge (and quickly) to develop this explanation, and to the extent that this paper can contribute to that body, it is important that the lack of correlation between present-day aerosol forcing and ECS is promulgated. Is it fair to say, then, that the mystery of CMIP6 ECS persists or, perhaps, deepens?

The paper notes that the spread in ERF between models is narrowed relative to CMIP5. This is a most welcome finding, given the poor specification of forcings in CMIP5. I recommend that the paper indicate that the result is consistent with a high-level recommendations from the Stouffer et al, 2017 paper (doi: 10.1175/BAMS-D-15-00013.1).

The narrowing of the range in aerosol forcing is particularly notable and welcome. That being said, the authors should point out in the abstract the importance of the aerosol forcing adjustments and large range in model results, especially with respect to clouds. The finding is included in the paper already and is notable in that it highlights challenges for the scientific community that studies aerosol-cloud interactions.

The limited importance of land use for forcing is surprising, and the spatial patterns there appears to be strong, with some overlap with aerosol forcing. Is there cancellation or reinforcement for these effects?

The final point made by the authors in the conclusion, which is that there is a need to constrain cloud responses to forcing since they contribute to the largest uncertainty in forcing, is well-taken but disturbing. Clouds appear to be not just a problem for

feedbacks, as is widely accepted by the community and has motivated a sustained focus on constraining cloud feedbacks, but they are a problem for forcing as well. This point should also be in the abstract and discussed in the abstract

However, the recommendation of the authors is vague and it is highly unclear to me how on how cloud responses can be constrained. Through process studies? Developing observational constraints? There are strikingly strong spatial patterns of ERF. Can some type of fingerprinting be used? The authors should indicate in the paper whether or not there even is a path forward for actually constraining these cloud responses or if the community needs to develop one before even being able to go down it to actually develop those constraints.

Minor points:

The x-axes on Figure 4 need fixing.

Figure 5 has lots of information but is confusing in there is concurrence between models in the spatial patterns of ERF but there appears to be little concurrence in some of the spatial patterns of adjustments and cloud contributions, even though when summed up, they are significant across models. This is even more the case for Figures 7 and 11, and some explanation of how this is achieved is needed for readers.

Line 386: Should be "equivalent" not "equalivent"
* * *

---

## Referee Comment (RC2) · Anonymous Referee #2 · 19 Mar 2020

This paper investigates the effective radiative forcing (ERF) from 13 CMIP6 models, and contributes to the RFMIP project. It presents contributions of particular climate forcers to anthropogenic forcing, including greenhouse gases, aerosols, and land-use. Results show a smaller anthropogenic ERF compared to AR5, and it is contributed by a stronger aerosol ERF. Additionally, the range of aerosol ERF from CMIP6 is narrower than CMIP5. This work introduces a range of methods to calculate ERF and adjustments as well. It is certainly a very comprehensive work and would make a valuable contribution to IPCC next assessment report. However, I feel there still could be some more interpretations of the work presented here. For these reasons, I am recommending this paper to be accepted for publications with minor revisions.

General comments:

1. It is interesting to see the range of aerosol ERF is narrower in CMIP6. However, it could be better if the authors can demonstrate which part (e.g., ERFari or ERFaci) contributes to the improvement most, and why is it.

2. The Introduction part could add some a short paragraph to talk about the contribution of aerosols, GHGs, and land-use to anthropogenic ERF, in terms of sign and magnitude. For example, how aerosols ERF counteracts a large part of the warming effect from GHGs meanwhile has the largest uncertainty.

3. Fig 8: Not fully understand why do the correlation between aerosol ERF and ECS/TCR. According to the definition, ECS and TCR are directly related to CO2, so it won't be a surprise to me that the correlation is bad. Can you give more explanations here?

4. This paper provides a number of methods to examine ERF from different climate forcers by using several climate models. It is certainly a very comprehensive work. However, it is easy to get lost when I am trying to understand the results. It would be of interest if some further work can be done to help the audience to better understand the results (not necessarily in this paper). For example, the adjustment from clouds contributes to most of the uncertainties. Are these uncertainties caused by different methods or different models? If it is caused by model variability, then what are the essential parameterization of clouds been used in these models? The geographical patterns shown in Fig 5, 6, 7, 11, and 12 are interesting, and it would be nice if the authors can explore more on them.

Specific comments:

1. Figure 2: maybe put this figure in the supplementary file? It is an interesting figure in terms of methodology, but not very necessarily related to the story and may distract readers.

2. Figure 3: GISS-E2-1-G is acting very differently to other models, especially on

adjustments from aerosols. Why is that?

3. Fig 4, x axis: It is hard to read the rightmost labels as they are overlaid.

4. Line 276 and the following paragraph: "This effect is traced to a slight cooling in the mid-troposphere in this model whereas other models show a distinct warming." It is interesting, but why GISS-E2-1-G shows a cooling which is apparently different from other models.

5. Line 331: "Atmospheric adjustments are small in magnitude in the aerosol forcing experiment, but large enough such that there is a noticeable difference between ERF and RF." I assume this conclusion is derived from table 5?

6. Line 338 and the following paragraph: I agree with the explanations. However, it is possible that absorbing aerosols play a minor role compared to non-absorbing aerosols just due to the smaller BC emissions than sulphate?

7. Line 410: Why LW ERFari+aci from the double call method doesn't always equal the total ERF? According to equation 8 and 9, it should be closed. And. Additionally, is this only for LW or both SW and LW?

8. Fig 11: I am a bit confused about land-use ERF results here. I can understand that it is small on global averages. However, I am surprised that it is still insignificant in some regions (e.g., North America, China), even though the regional ERF there is large ($\sim$ -6 W m-2) (Fig 11). How's the significance been calculated?

---

## Author Comment (AC1) · 20 May 2020

Anonymous Referee #1:

General Comments:

The paper presents a comprehensive analysis of the diagnosed values of effective radiative forcing (ERF) for the CMIP6 models, and breaks down the contributions of this forcing from greenhouse gases, aerosols, and land-use. The use of ERF has continued to grow and it is now at least as widely-used, if not more so, than traditional metrics of forcing such as instantaneous forcing or stratospheric-adjusted forcing. It is an important paper for benchmarking the performance of CMIP6 models, and its findings will hopefully be used in upcoming assessment reports. That being said, there

are many different findings in this paper and it would likely be more digestible if the findings were explored in more detail in separate papers. This is something of an omnibus paper. However, despite its size, the paper has important findings and only requires minor revisions before being acceptable for publication.

» Thank you for your positive comments on the paper. We agree that it is a long paper and we appreciate your time, energy and effort given to the thorough review.

» It is always a consideraton whether to split papers such as this into more than one volume. We were of the belief that just reporting the headline fixed-SST ERF results from the CMIP6 ensemble "as-is" did not constitute enough analysis to merit a paper on its own, although these figures (table 2) will possibly be the most widely-used. I felt that Smith et al. (2018b) would have benefitted from the regional analysis, so I was keen to include it in this submission. It sounds like this motivated interesting questions from both reviewers, so was a useful addition, at the expense of a longer paper.

Specific Comments:

The last point made by the authors in the abstract, which appears to be supported in Figure 8, namely that they see no evidence that aerosols are contributing to the spread in ECS, is striking and bears more discussing. The range and change in ECS for CMIP6 is and should be of great concern to the large numbers of individuals involved in CMIP6 and to the scientific community as a whole, since an explanation is required. I hope that a body of literature will emerge (and quickly) to develop this explanation, and to the extent that this paper can contribute to that body, it is important that the lack of correlation between present-day aerosol forcing and ECS is promulgated. Is it fair to say, then, that the mystery of CMIP6 ECS persists or, perhaps, deepens?

» Thank you for picking out this key point which we could highlight more. The last sentence of the abstract has been extended:

» "Therefore, there is no evidence to suggest that the increasing spread in climate

sensitivity in CMIP6 models, particularly related to high-sensitivity models, is a conse-
quence of a stronger negative present-day aerosol forcing, and little evidence that mod-
elling groups are systematically tuning climate sensitivity or aerosol forcing to recreate
observed historical warming."

» From the model description papers cited in table 1, with two exceptions (MPI-ESM1-
2 and EC-Earth3), there is either an explicit mention that historical temperatures were
not used as a model performance indicator for their CMIP6 configuration, or the model
paper was silent on this. Indeed it is evident from a number of models' historical tem-
perature evolutions that they were not a target of model tuning:

- CanESM5 (high climate sensitivity, moderately low present-day aerosol forcing)
shows more warming than observations over the historical period (Swart et al., 2019);

- NorESM2-LM and NorESM2-MM (lower Gregory sensitivity, stronger aerosol forcing)
shows less warming than observed (Seland et al., 2020). (The NorESM2 models ac-
tually have high equilibrium sensitivity but low sensitivity as measured from a 150-year
Gregory regression due to strongly increasing feedbacks over time).

- UKESM1-0-LL (high sensitivity, about average aerosol forcing) has approximately the
correct level of present-day warming but is too cool in the 1960-2000 period (Sellar et
al., 2019).

» A footnote was added in section 5.3.2:

» MPI-ESM1-2 (Mauritsen et al., 2019) and EC-Earth3 (Wyser et al., 2019) are the only
documented exceptions. MIROC6 (Tatebe et al., 2019) did tune the aerosol forcing
to better correspond to the AR5 best estimate but explicitly did not tune for surface
temperature.

» While the magnitude of year-2014 aerosol ERF may not constrain climate sensitivity,
potentially the evolution of aerosol forcing since 1970 may provide some constraint on
transient climate response. As different models include different aerosol processes,

the time history of aerosol forcing can be quite different in different models even when driven with the same emissions. 6 models so far have performed the RFMIP Tier 2 aerosol forcing transient experiment and future work will investigate this.

The paper notes that the spread in ERF between models is narrowed relative to CMIP5. This is a most welcome finding, given the poor specification of forcings in CMIP5. I recommend that the paper indicate that the result is consistent with a high-level recommendations from the Stouffer et al, 2017 paper (doi: 10.1175/BAMS-D-15-00013.1).

» Thank you for the suggestion. In the Conclusion (line 491) we add a sentence:

» "This has helped to address a concern from CMIP5: that forcing was poorly characterised in CMIP5 models and inconsistently determined (Stouffer et al., 2017)."

» We should caveat, and we discuss later in the paragraph, that there are more models in CMIP6 that did not submit the RFMIP Tier 1 experiments in which aerosol forcing is probably stronger than the lower bound from the 17 models for which we have data:

» "Although 17 models is a reasonable sample size of the CMIP6 population, more models may submit forcing results to CMIP6 that would widen this range (and indeed, we would encourage modelling groups to do so). One example is E3SM which did not perform the RFMIP aerosol forcing experiment but where it would be likely that the 1850–2014 aerosol forcing would be more negative than -1.37 W m-2 (fig. 25 in Golaz et al., 2019)."

» This may also have been true in CMIP5 in which the sstClimAerosol experiment was from a subset of models.

The narrowing of the range in aerosol forcing is particularly notable and welcome. That being said, the authors should point out in the abstract the importance of the aerosol forcing adjustments and large range in model results, especially with respect to clouds. The finding is included in the paper already and is notable in that it highlights challenges for the scientific community that studies aerosol-cloud interactions.

[Figure]

» In combination with one of your comments further below, this sentence has been added to the abstract:

» "In most cases, the largest contributors to the spread in ERF is from the instantaneous radiative forcing (IRF) and from cloud responses, particularly aerosol-cloud interactions to aerosol forcing."

» Along with the previous comment, in order to more clearly illustrate comparisons with CMIP5 which were discussed in the text but not graphically compared in the first submission, a new Figure 5 has been added which detail these comparisons, clearly showing the increased 4xCO2 forcing in CMIP6 and slightly reduced range of aerosol forcing.

The limited importance of land use for forcing is surprising, and the spatial patterns there appears to be strong, with some overlap with aerosol forcing. Is there cancellation or reinforcement for these effects?

» To attempt to explain this further, the change in aerosol optical depth in the nine models that output this diagnostic in the land-use experiment is shown in fig. S4. Aerosol-induced changes in some models are notable but not large. For example CanESM5 has the greatest increase in aerosol optical depth over the Northern Hemisphere land regions but is relatively weak in terms of land use forcing. The exception to this is for the models that include ice nucleation effects from biogenic aerosol which is coupled to the land surface scheme. This is clearly the case for NorESM2-LM, where the cloud adjustment dominates the forcing and results in a positive land-use change ERF, and is already documented in fig. S3. All of the other 13 models have a negative land use forcing, which are in line with an observationally-constrained estimate from CMIP5 models (Lejeune et al., 2020).

» As you assert, regionally land-use change is important. In regions experiencing lots of deforestation (North America, Western Eurasia and South America) albedo is increased causing a negative ERF (fig. 12). In these regions land-use forcing determines the multi-model mean ERF, however, while models agree on the negative land-use ERF, there is no model consensus on the net forcing (fig. 13). Compared to land-use change, aerosols cause a slightly weaker negative forcing in the deforested regions, which in the northern hemisphere is caused by an increase in SW reflectance (fig. 8). The aerosol effect will reduce downward SW at the surface and so reduce the effect of surface albedo changes caused by land-use change. The cause of the negative aerosol ERF in the South American deforested region is more complex and here there is no obvious non-linearity in the combined land-use and aerosol ERF.

» It should also be noted that land-use change does not only impact climate via radiative forcing and that the temperature impacts of other mechanisms are often larger and of opposite sign (e.g. Bright et al., 2017).

The final point made by the authors in the conclusion, which is that there is a need to constrain cloud responses to forcing since they contribute to the largest uncertainty in forcing, is well-taken but disturbing. Clouds appear to be not just a problem for feedbacks, as is widely accepted by the community and has motivated a sustained focus on constraining cloud feedbacks, but they are a problem for forcing as well. This point should also be in the abstract and discussed in the abstract.

» From tables 3, 4, 5, 7 and 8 it is apparent the largest contributions to the spread in ERF are from IRF and cloud adjustments. In addition to improving cloud processes, there is still some way to go in radiative transfer modelling. Work is in progress under RFMIP to do this, and we show in Figure 5 for 4xCO2 that the inter-model spread is much reduced compared to CMIP5 which may be indicative of an improvement in model radiative transfer.

» In conclusion, added:

» "The instantaneous radiative forcing and cloud adjustments are generally the largest sources of inter-model spread in the forcing component in climate models. Since IRF is not directly calculated in this study, some of this spread may be from residuals in the

kernel decomposition and the true spread in IRF may be smaller than reported here. One strand of RFMIP will include benchmarking of GCM radiative transfer against line-by-line codes. Radiative transfer is a well-grounded theoretical problem where the diversity in line-by-line codes is small (Pincus et al., 2015), so this component of inter-model diversity has a measurable yardstick for improvement."

However, the recommendation of the authors is vague and it is highly unclear to me how on how cloud responses can be constrained. Through process studies? Developing observational constraints? There are strikingly strong spatial patterns of ERF. Can some type of fingerprinting be used? The authors should indicate in the paper whether or not there even is a path forward for actually constraining these cloud responses or if the community needs to develop one before even being able to go down it to actually develop those constraints.

» We agree that the last sentence of the conclusion was vague in the first submission. We have added some suggestions at the end of the conclusion (following on from the passage above) and a final summary sentence that links back to the introduction.

» "Cloud responses are more difficult to constrain and exhibit a wide range of behaviour to both greenhouse gas and aerosol forcing. However, progress is beginning to be made. For greenhouse gas forcing, techniques from the climate feedback literature that have observational parallels, such as analysing cloud-controlling factors (Klein et al., 2017), can be applied to adjustments. Use of the ISCCP simulator diagnostics with the ISCCP cloud kernel, another method conceptualised by climate feedback investigations (Zelinka et al., 2012), allows cloud adjustments to be calculated directly facilitating better inter-model comparison. For aerosol forcing, observational methods exist to determine RFari and RFaci using satellite and reanalysis data (Bellouin et al., 2013; Bellouin et al., 2020a). Ultimately, reducing uncertainty in effective radiative forcing will reduce uncertainty in climate projections due to the central role of forcing in driving the Earth's global mean temperature response."

Minor points:

The x-axes on Figure 4 need fixing.

» Rotated x-axis labels to make clearer.

Figure 5 has lots of information but is confusing in there is concurrence between models in the spatial patterns of ERF but there appears to be little concurrence in some of the spatial patterns of adjustments and cloud contributions, even though when summed up, they are significant across models. This is even more the case for Figures 7 and 11, and some explanation of how this is achieved is needed for readers.

» You make a good point here. Forcing adjustments are in many cases robust in sign in the global mean change but less so spatially between models. This highlights the point that forcing and adjustments are best considered globally averaged quantities. At the end of section 5 (line 481) we have added the following:

» "For all forcings, but particularly for land-use, aerosol and total anthropogenic, many of the forcing and adjustment terms do not show robust signals regionally. This indicates that adjustments are best considered as global-mean quantities that affect the globally-resolved forcing-feedback framework (Eq (1))."

Line 386: Should be "equivalent" not "equalivent"

» Typo corrected - thank you.

» References used in response to reviewer #1 not in manuscript: Bright, R. M., Davin, E., O'Halloran, T., Pongratz, J., Zhao, K. and A. Cescatti, 2017: Local temperature response to land cover and management change driven by non-radiative processes. Nat. Climate Change, 7, 296-302, https://doi.org/10.1038/nclimate3250.

---

## Author Comment (AC2) · 20 May 2020

Anonymous Referee #2:

This paper investigates the effective radiative forcing (ERF) from 13 CMIP6 models, and contributes to the RFMIP project. It presents contributions of particular climate forcers to anthropogenic forcing, including greenhouse gases, aerosols, and land-use. Results show a smaller anthropogenic ERF compared to AR5, and it is contributed by a stronger aerosol ERF. Additionally, the range of aerosol ERF from CMIP6 is narrower than CMIP5. This work introduces a range of methods to calculate ERF and adjustments as well. It is certainly a very comprehensive work and would make a valuable contribution to IPCC next assessment report. However, I feel there still could be some

more interpretations of the work presented here. For these reasons, I am recommending this paper to be accepted for publications with minor revisions.

» Thank you for your thorough review of this paper and positive comments. We try to address the further interpretations that you mention in the comments below.

General comments:

1. It is interesting to see the range of aerosol ERF is narrower in CMIP6. However, it could be better if the authors can demonstrate which part (e.g., ERFari or ERFaci) contributes to the improvement most, and why is it.

» Table 6 shows the ERFari and ERFaci components, which can be compared with Zelinka et al. (2014) for CMIP5 models. The standard deviation of model estimates has reduced for ERFari, ERFaci and ERFari+aci (0.19 W m-2, 0.30 W m-2 and 0.20 W m-2 respectively compared to 0.22 W m-2, 0.34 W m-2 and 0.30 W m-2 for CMIP5 in Zelinka et al. (2014)). The reduced standard deviation is a consequence of a slightly smaller spread in each of the components with more models. It is not clear that one component of the aerosol forcing has shown a greater reduction in spread than the other between CMIP5 and and CMIP6. However, in aid of easier comparison of the total aerosol forcing (and CO2), we have included a new figure 5 that compares CMIP5 and CMIP6.

» We have been careful to point out that the models submitting results to Tier 1 of RFMIP are only a subset of all CMIP6 models, and inclusion of more models could extend the range of aerosol ERF. In the conclusion we have added:

» "Although 17 models is a reasonable sample size of the CMIP6 population, more models may submit forcing results to CMIP6 that would widen this range (and indeed, we would encourage modelling groups to do so). One example is E3SM which did not perform the RFMIP aerosol forcing experiment but where it would be likely that the 1850–2014 aerosol forcing would be more negative than -1.37 W m-2 (fig. 25 in Golaz

et al., 2019)."

» This may also have been true in CMIP5 in which the sstClimAerosol experiment was from a subset of 10 models.

2. The Introduction part could add some a short paragraph to talk about the contribution of aerosols, GHGs, and land-use to anthropogenic ERF, in terms of sign and magnitude. For example, how aerosols ERF counteracts a large part of the warming effect from GHGs meanwhile has the largest uncertainty.

» We agree that this would give the study some additional context. At the end of the first paragraph we have included the following:

» "Since the start of the Industrial Era until the present-day, anthropogenic forcing has typically been increasing, and has been the dominant component of the total forcing on the Earth system except for brief periods following large volcanic eruptions (Myhre et al., 2013). The main constituents of anthropogenic ERF are a positive forcing from greenhouse gases and a partially offsetting negative forcing from aerosols. While greenhouse gas forcing is reasonably well-known, aerosol forcing is more uncertain due to the spatial variation of aerosols, their short atmospheric lifetime, and their complex interactions with clouds (Boucher et al., 2013; Bellouin et al., 2020b)."

3. Fig 8: Not fully understand why do the correlation between aerosol ERF and ECS/TCR. According to the definition, ECS and TCR are directly related to CO2, so it won't be a surprise to me that the correlation is bad. Can you give more explanations here?

» One goal of researchers analysing CMIP6 models is to try and understand the drivers of increased climate sensitivity compared to CMIP5, and whether the very high sensitivity models are realistic and under what circumstances. If realistic, this means that high sensitivity cannot be ruled out, which is an important result for policy communication. In order to reproduce historically-observed warming a high ECS and high

TCR requires a strongly negative aerosol forcing. I wasn't the first to do this but this is shown in an energy balance framework with a large probabilistic ensemble in Figure 7 of Smith et al. (2018a). In coupled models, Kiehl (2007) showed that ECS and aerosol forcing were negatively correlated in CMIP3, as well as in CMIP5 for models that included an aerosol-cloud interaction (Chylek et al., 2016) but not for the model population as a whole (Forster et al., 2013). The lack of correlation in CMIP6 suggests that climate modelling groups are not using historical warming observations as a model tuning constraint, an assertion that is on the whole verified from a review of available CMIP6 model description papers listed in table 1 except for the two counterexamples specifically noted (footnote on page 22). It does mean we cannot constrain climate sensitivity using model-diagnosed aerosol forcing in the present day.

» You touch on a relevant point here on the contribution of CO2 forcing to climate sensitivity. As Zelinka et al. (2020) showed, the increase in (150-year Gregory) CO2 forcing is a small but substantial contributor to the increase in climate sensitivity. We also show this to be the case in the fixed-sea surface temperature definition of ERF and this is highlighted in Figure 5.

4. This paper provides a number of methods to examine ERF from different climate forcers by using several climate models. It is certainly a very comprehensive work. However, it is easy to get lost when I am trying to understand the results. It would be of interest if some further work can be done to help the audience to better understand the results (not necessarily in this paper). For example, the adjustment from clouds contributes to most of the uncertainties. Are these uncertainties caused by different methods or different models? If it is caused by model variability, then what are the essential parameterization of clouds been used in these models? The geographical patterns shown in Fig 5, 6, 7, 11, and 12 are interesting, and it would be nice if the authors can explore more on them.

» Addressing the point of ERF methods first, there is some discussion as to how to best define effective radiative forcing with the goal that it should be a convertible currency

for measuring long-term surface temperature changes from different forcers. This was motivated originally by Hansen et al. (2005) and discussed recently in Tang et al. (2019) and Richardson et al. (2019), where correcting for the land surface warming improves the ERF to temperature relationship slightly, so we provide the alternatives in Figure 1. Although it extends the discussion slightly, it puts these alternative ERF definitions in the open, with the actual numbers in new supplementary tables for interested readers. We focus the rest of the paper on the fixed SST results, one reason being the separation into adjustments is difficult or ill-defined for some of the other methods.

» Addressing the point on cloud uncertainties, this is a very good suggestion for future work. One question that was raised when I presented this at the virtual EGU conference was whether there was any relationship between the complexity of cloud parameterisation and forcing, and it is one that we do not know the answer to. For climate sensitivity results, a more realistic cloud water phase parameterisation has led to an increase in sensitivity in CESM2 (Gettleman et al., 2019), and I believe also UKESM1-0-LL (compared to HadGEM2-ES). It would be interesting to determine whether this also applies to adjustments. Combining this with our response to Reviewer #1, we have added some more explanation near the end of the discussion around how cloud changes could possibly be constrained:

» "Cloud responses are more difficult to constrain and exhibit a wide range of behaviour to both greenhouse gas and aerosol forcing. However, progress is beginning to be made. For greenhouse gas forcing, techniques from the climate feedback literature that have observational parallels, such as analysing cloud-controlling factors (Klein et al., 2017), can be applied to adjustments. Use of the ISCCP simulator diagnostics with the ISCCP cloud kernel, another method conceptualised by climate feedback investigations (Zelinka et al., 2012), allows cloud adjustments to be calculated directly facilitating better inter-model comparison. For aerosol forcing, observational methods exist to determine RFari and RFaci using satellite and reanalysis data (Bellouin et al., 2013; Bellouin et al., 2020a)."

Specific comments:

1. Figure 2: maybe put this figure in the supplementary file? It is an interesting figure in terms of methodology, but not very necessarily related to the story and may distract readers.

» We can see arguments for and against moving this figure to the Supplement and agree with the reviewer that it is probably not central to the discussion of forcing. However, we believe that it is an important point to highlight to others. We have already in fact obtained one citation for this paper already by pointing out that the forcing behaviour of CNRM-ESM2-1 is different to other models (Williams, Ceppi & Katavouta, accepted in Environmental Research Letters), so believe it is a useful reference for the community.

2. Figure 3: GISS-E2-1-G is acting very differently to other models, especially on adjustments from aerosols. Why is that?

» An error was discovered in the first version on ESGF of the GISS-E2-1-G 4xCO2 simulation. This has now been corrected on ESGF and the ERF and adjustments have changed for this model and in line with other 4xCO2 experiments. Several parts of the text discussing the old treatment have been modified or deleted.

3. Fig 4, x axis: It is hard to read the rightmost labels as they are overlaid.

» Rotated x-axis labels to make distinction clearer.

4. Line 276 and the following paragraph: "This effect is traced to a slight cooling in the mid-troposphere in this model whereas other models show a distinct warming." It is interesting, but why GISS-E2-1-G shows a cooling which is apparently different from other models.

» As mentioned in the response to point 2 above, an error in the GISS model was corrected and this tropospheric cooling no longer exists. This paragraph has been deleted.

5. Line 331: "Atmospheric adjustments are small in magnitude in the aerosol forcing experiment, but large enough such that there is a noticeable difference between ERF and RF." I assume this conclusion is derived from table 5?

» This follows from table 5 (as the sum of IRF and ta_st) and figure 1. The table does not list RF separately. This has now been done for all experiments, and added to the Supplement as tables S1 to S5. A reference to figure 1 and the supplementary table S3 has been inserted.

6. Line 338 and the following paragraph: I agree with the explanations. However, it is possible that absorbing aerosols play a minor role compared to non-absorbing aerosols just due to the smaller BC emissions than sulphate?

» Good point, also shown by the magnitude of the individual forcings in AerChemMIP (Thornhill et al., 2020). The text has been updated as follows:

» "This also implies that absorbing aerosols play only a minor role in most models, as BC induces strong adjustments that cause a general increase in cloud height in PDRMIP models from an increasing tropospheric stability (Smith et al. (2018b); Stjern et al. (2017); fig. S2). There is no evidence of this in the RFMIP aerosol forcing experiment, although some models do also include aerosol-cloud interactions from BC, and the effect may be due to the BC forcing being a smaller fraction of the total aerosol forcing than sulfate (Thornhill et al., 2020)."

7. Line 410: Why LW ERFari+aci from the double call method doesn't always equal the total ERF? According to equation 8 and 9, it should be closed. And. Additionally, is this only for LW or both SW and LW?

» This section has been simplified a little. Going back to the original Ghan (2013) reference for the aerosol forcing double call, the aerosol ERF can be broken down into direct radiative forcing (RFari in AR5 terminology), cloud radiative forcing (ERFaci plus the semi-direct effect AKA the adjustment part of ERFari) and surface albedo adjust-

ment. In this case, the breakdown given in our paper has slightly incorrect terminology (see equations below) but the same bias exists as for APRP as discussed in Zelinka et al. (2014) which is why these methods are compared in Figure 11. Because the semi-direct is small due to the small influence of black carbon in these results, any adjustments to RFari that are being counted as part of ERFaci will not affect the results too much. Spelling out these equations for the double call with reference to Ghan (2013) and focusing on the SW:

» Direct radiative forcing (RFari) = $-\Delta$rsut - ($-\Delta$rsutaf)

» Cloud radiative forcing (ERFaci + semi-direct effect) = $-\Delta$rsutaf - ($-\Delta$rsutcsaf)

» Surface albedo forcing = $-\Delta$rsutcsaf

» Summing up the three components, everything except $-\Delta$rsut cancels out, which is identically the SW ERF. The same breakdown is done for the LW. So you are correct that the full breakdown should be exact.

» By reporting ERFari+aci, we neglect the surface albedo component, which could usefully be compared to the surface albedo adjustment calculated by the kernel method (although we do not do so). The "surface albedo" forcing is not zero in the LW decomposition using the double call either, which may be compared to the surface temperature or other LW adjustments.

» As an additional confidence that our results are correctly calculated, they agree with results obtained for CNRM-ESM2-1 and CNRM-CM6-1 independently by Séférian et al. (2019) and Michou et al. (2020).

8. Fig 11: I am a bit confused about land-use ERF results here. I can understand that it is small on global averages. However, I am surprised that it is still insignificant in some regions (e.g., North America, China), even though the regional ERF there is large (-6 W m-2) (Fig 11). How's the significance been calculated?

» Originally, significance in these plots was defined as the multi-model mean being

different from zero at the one standard-deviation level in each grid cell. In the case of land use, the standard deviation can be quite large in some of the North American grid cells (and larger than the absolute mean) as some models show strong negative ERF and some show weak negative ERF, but the ERF is still robustly negative. This was, probably incorrectly, leading to a lack of "significance" under this definition. For this reason, and to be consistent with the ISCCP simulator plot in fig. 4, the definition is now that 75% or more of models should agree on sign. In addition, the hatching scheme on these figures has been improved so it should be clearer which regions are shaded out.

» In the caption to fig. 7, added

» "Hatched regions are where less than 75% of models agree on the sign of the change."

» Where spatial plots first introduced in the text in line 300, added

» "Hatched areas are defined where less than 75% of models agree on the sign of the change."

»References used in response to reviewer #2 not in manuscript:

»Michou, M., Nabat, P., Saint-Martin, D., Bock, J., Decharme, B., Mallet, M., et al. ( 2020). Present-day and historical aerosol and ozone characteristics in CNRM CMIP6 simulations. Journal of Advances in Modeling Earth Systems, 12, e2019MS001816. https://doi.org/10.1029/2019MS001816

»Gettelman, A., Hannay, C., Bacmeister, J. T., Neale, R. B., Pendergrass, A. G., Danabasoglu, G., et al. (2019). High climate sensitivity in the Community Earth System Model Version 2 (CESM2). Geophysical Research Letters, 46, 8329– 8337. https://doi.org/10.1029/2019GL083978